# Stochastic Gradient Hamiltonian Monte Carlo Methods with Recursive Variance Reduction

**Difan Zou**
Department of Computer Science
University of California, Los Angeles
Los Angeles, CA 90095
knowzou@cs.ucla.edu

**Pan Xu**
Department of Computer Science
University of California, Los Angeles
Los Angeles, CA 90095
panxu@cs.ucla.edu

**Quanquan Gu**
Department of Computer Science
University of California, Los Angeles
Los Angeles, CA 90095
qgu@cs.ucla.edu

## Abstract

Stochastic Gradient Hamiltonian Monte Carlo (SGHMC) algorithms have received increasing attention in both theory and practice. In this paper, we propose a Stochastic Recursive Variance-Reduced gradient HMC (SRVR-HMC) algorithm. It makes use of a semi-stochastic gradient estimator that recursively accumulates the gradient information to reduce the variance of the stochastic gradient. We provide a convergence analysis of SRVR-HMC for sampling from a class of non-log-concave distributions and show that SRVR-HMC converges faster than all existing HMC-type algorithms based on underdamped Langevin dynamics. Thorough experiments on synthetic and real-world datasets validate our theory and demonstrate the superiority of SRVR-HMC.

## 1 Introduction

Monte Carlo Markov Chain (MCMC) has been widely used in Bayesian learning [1] as a powerful tool for posterior sampling, inference and decision making. More recently, Hamiltonian MCMC approaches based on the Hamiltonian Langevin dynamics [24, 43] have received extensive attention in both theory and practice [16, 5, 40, 14, 6, 18, 55, 28] due to their widespread empirical successes. Hamiltonian Langevin dynamics (a.k.a., underdamped Langevin dynamics) [19] is described by the following stochastic differential equation:

$$
\begin{aligned}
\mathrm{d}\boldsymbol{V}_t &= -\gamma \boldsymbol{V}_t \mathrm{d}t - u\nabla f(\boldsymbol{X}_t)\mathrm{d}t + \sqrt{2\gamma u}\mathrm{d}\boldsymbol{B}_t, \\
\mathrm{d}\boldsymbol{X}_t &= \boldsymbol{V}_t \mathrm{d}t,
\end{aligned}
\tag{1.1}
$$

where $\gamma > 0$ is called the friction parameter, $u > 0$ is the inverse mass, $\boldsymbol{X}_t, \boldsymbol{V}_t \in \mathbb{R}^d$ are the position and velocity variables of the continuous-time dynamics respectively, and $\boldsymbol{B}_t \in \mathbb{R}^d$ is the standard Brownian motion. Under mild assumptions on the function $f(\mathbf{x})$, the Markov process $(\boldsymbol{X}_t, \boldsymbol{V}_t)$ has a unique stationary distribution which is proportional to $\exp\{-f(\mathbf{x}) - \|\mathbf{v}\|_2^2/(2u)\}$ and the marginal distribution of $\boldsymbol{X}_t$ converges to a stationary distribution $\pi \propto \exp\{-f(\mathbf{x})\}$. Hence, we can apply numerical integrators to discretize the continuous-time dynamics (1.1) in order to sample from the target distribution $\pi$. Direct Euler-Maruyama discretization [34] of (1.1) gives rise to

$$
\begin{aligned}
\mathbf{v}_{k+1} &= \mathbf{v}_k - \gamma\eta\mathbf{v}_k - \eta u\nabla f(\mathbf{x}_k) + \sqrt{2\gamma u\eta}\epsilon_k, \\
\mathbf{x}_{k+1} &= \mathbf{x}_k + \eta\mathbf{v}_k,
\end{aligned}
\tag{1.2}
$$

which is known as underdamped Langevin MCMC (UL-MCMC) and can also be viewed as a type of Hamiltonian Monte Carlo (HMC) methods [43, 6]. Cheng et al. [18] studied a modified version of UL-MCMC in (1.2) and proved its convergence rate to the stationary distribution in 2-Wasserstein distance for sampling from strongly log-concave densities. When the target distribution is non-log-concave but admits certain good properties, the convergence guarantees of UL-MCMC in Wasserstein metric have also been established in [27, 17, 8, 30].

In practice, $f(\mathbf{x})$ in (1.2) can be chosen as the negative log-likelihood function on the training data:

$$f(\mathbf{x}) = n^{-1} \sum_{i=1}^{n} f_i(\mathbf{x}), \tag{1.3}$$

where $n$ is the size of training data and $f_i(\mathbf{x}) : \mathbb{R}^d \to \mathbb{R}$ is the negative log-likelihood function on the $i$-th data point. For a large dataset, it can be extremely inefficient to compute the full gradient $\nabla f(\mathbf{x})$ which consists of gradients $\nabla f_i(\mathbf{x})$'s for all data points. To alleviate this computational burden, stochastic gradient Hamiltonian Monte Carlo (SGHMC) methods [16, 40] and stochastic gradient UL-MCMC (SG-UL-MCMC) [18] were proposed, which replace the full gradient in (1.2) with a mini-batch stochastic gradient. While SGHMC is much more efficient than HMC methods, it comes at the cost of a slower mixing rate due to the large variance caused by stochastic gradients [5, 6, 23]. To resolve this dilemma, Zou et al. [55], Li et al. [37] proposed stochastic variance-reduced gradient HMC methods using variance reduction techniques [33, 36] and proved that variance reduction can accelerate the convergence of both HMC and SGHMC for sampling and Bayesian inference. For sampling from a class of non-log-concave densities, Gao et al. [30] showed that SGHMC converges to the stationary distribution of (1.1) up to an $\epsilon$-error in 2-Wasserstein distance with $\widetilde{O}(\epsilon^{-8}\mu_*^{-5})$[1] gradient complexity[2], where $\mu_*$ is a lower bound of the spectral gap of the Markov process generated by (1.1) and is in the order of $\exp(-\widetilde{O}(d))$ in the worst case [27]. This gradient complexity of SGHMC is very high even for a moderate sampling error $\epsilon$.

In this paper, we aim to reduce the gradient complexity of SGHMC for sampling from non-log-concave densities. The fundamental challenge in speeding up HMC-type methods lies in the control of the discretization error between the Hamiltonian Langevin dynamics (1.1) and discrete algorithms. We propose a novel algorithm, namely stochastic recursive variance-reduced gradient HMC (SRVR-HMC), which employs a recursively updated semi-stochastic gradient estimator to reduce the variance of stochastic gradient and improve the discretization error. Note that such a recursively updated semi-stochastic gradient estimator was originally proposed in [44, 29] for finding stationary points in stochastic nonconvex optimization. Nevertheless, our analysis is fundamentally different from that in [44, 29] since their goal is just to find a stationary point of $f(\mathbf{x})$, while we aim to sample from the target distribution $\pi \propto \exp(-f(\mathbf{x}))$ that concentrates on the global minimizer of $f(\mathbf{x})$, which is substantially more challenging.

## 1.1 Our contributions

We summarize our major contributions as follows.

- We propose a new HMC algorithm called SRVR-HMC for approximate sampling, which is built on a recursively updated semi-stochastic gradient estimator that significantly decreases the discretization error and speeds up the sampling process.

- We establish the convergence guarantee of SRVR-HMC for sampling from non-log-concave densities satisfying certain dissipativeness condition. Specifically, we show that its gradient complexity for achieving $\epsilon$-error in 2-Wasserstein distance is $\widetilde{O}((n + \epsilon^{-2}n^{1/2}\mu_*^{-3/2}) \wedge \epsilon^{-4}\mu_*^{-2})$. Remarkably, the convergence guarantee of SRVR-HMC is better than the $\widetilde{O}(\epsilon^{-4}\mu_*^{-3}n)$ gradient complexity of HMC [30] by a factor of at least $\widetilde{O}(\epsilon^{-2}\mu_*^{-3/2}n^{1/2})$, and better than the $\widetilde{O}(\epsilon^{-8}\mu_*^{-5})$ gradient complexity of SGHMC [30] by a factor of at least $\widetilde{O}(\epsilon^{-4}\mu_*^{-3})$.

- With a proper choice of parameters, our algorithm can reduce to UL-MCMC [18] and SG-UL-MCMC [18], which are originally proposed for sampling from strongly-log-concave distributions.

Our theoretical analysis shows that these two algorithms can be used for sampling from non-log-concave distributions as well, and they enjoy lower gradient complexities than HMC and SGHMC [30], which is of independent interest.

- We compare our algorithm with many state-of-the-art baselines through experiments on sampling from Gaussian mixture distributions, independent component analysis (ICA) and Bayesian logistic regression, which further validates the superiority of our algorithm.

## 1.2 Additional related work

There is also a vast literature of MCMC methods based on the overdamped Langevin dynamics [35]:

$$\mathrm{d}\boldsymbol{X}_t = -\nabla f(\boldsymbol{X}_t)\mathrm{d}t + \sqrt{2\beta}\mathrm{d}\boldsymbol{B}_t, \tag{1.4}$$

where $\beta > 0$ is the temperature parameter and $\boldsymbol{B}_t$ is Brownian motion. The convergence analysis of Langevin based algorithms dates back to [46]. Mattingly et al. [41] established convergence rates for a class of discrete approximation of Langevin dynamics. When the target distribution is smooth and strongly log-concave, the convergence of Langevin Monte Carlo (LMC) based on the discretization of (1.4) has been widely studied in terms of both total variation (TV) distance [21, 26] and 2-Wasserstein distance [22, 20]. Welling and Teh [50] proposed the stochastic gradient Langevin dynamics (SGLD) algorithm to avoid full gradient computation. Teh et al. [47] proposed to apply decreasing step size with SGLD and proved its convergence in terms of mean square error (MSE). Vollmer et al. [48] characterized the bias of SGLD and further proposed a modified SGLD algorithm that removes the bias. [10] establish a link between LMC, SGLD, SGLDFP (a variant of SGLD) and SGD, which shows that the stationary distribution of LMC and SGLDFP can be closer to the target density $\pi$ as the sample size increases, while the dynamics of SGLD is more similar to that of SGD. Barkhagen et al. [4], Chau et al. [13] studied the convergence of SGLD when the training data in (1.3) are dependent. In order to reduce the variance of SGLD, SVRG-LD and SAGA-LD have been proposed by Dubey et al. [25] and their convergence have been studied in terms of MSE [25, 15] and 2-Wasserstein distance [56, 12]. Baker et al. [2] proposed to use control variate in SGLD which can also reduce the variance and improve the convergence rate. Mou et al. [42] studied the generalization performance of SGLD from both stability and PAC-Bayesian perspectives. For nonconvex optimization, Raginsky et al. [45] proved the non-asymptotic convergence rate of SGLD and Zhang et al. [52] analyzed the hitting time of SGLD to local minima. Xu et al. [51] further studied the global convergence of a class of Langevin dynamics based algorithms.

Table 1: Gradient complexity of different methods to achieve $\epsilon$-error in 2-Wasserstein distance for sampling from non-log-concave densities.

| Methods | Gradient Complexity | |
|---|---|---|
| LMC | $\widetilde{O}\big(\epsilon^{-4}\lambda_*^{-5}n\big)$ | [45] |
| SGLD | $\widetilde{O}\big(\epsilon^{-8}\lambda_*^{-9}\big)$ | [45] |
| SVRG-LD | $\widetilde{O}\big(n + \epsilon^{-2}\lambda_*^{-4}n^{3/4} + \epsilon^{-4}\lambda_*^{-4}n^{1/2}\big)$ | [57] |
| HMC | $\widetilde{O}\big(\epsilon^{-4}\mu_*^{-3}n\big)$ | [30] |
| UL-MCMC | $\widetilde{O}\big(\epsilon^{-2}\mu_*^{-3/2}n\big)$ | ▷ Corollary 3.9 |
| SGHMC | $\widetilde{O}\big(\epsilon^{-8}\mu_*^{-5}\big)$ | [30] |
| SG-UL-MCMC | $\widetilde{O}\big(\epsilon^{-6}\mu_*^{-5/2}\big)$ | ▷ Corollary 3.9 |
| **SRVR-HMC** | $\widetilde{O}\big((n + \epsilon^{-2}n^{1/2}\mu_*^{-3/2}) \wedge \epsilon^{-4}\mu_*^{-2}\big)$ | ▷ Corollary 3.5 |

In Table 1, we compare the gradient complexity of different methods to achieve $\epsilon$-error in 2-Wasserstein distance for sampling from non-log-concave densities[3]. LMC, SGLD and SVRG-LD are based on overdamped Langevin dynamics (1.4) and HMC, UL-MCMC, SGHMC, SG-UL-MCMC and SRVR-HMC are based on underdamped Langevin dynamics (1.1). The HMC/SGHMC algorithm studied in [30] and the UL-MCMC/SG-UL-MCMC algorithm [18] analyzed in this paper are

slightly different since they rely on different discretization methods to the Hamiltonian Langevin dynamics (1.1). In addition, note that $\lambda_*$ denotes the spectral gap of the Markov process generated by overdamped Langevin dynamics (1.4), which is also in the order of $\exp(-\widetilde{O}(d))$ [9, 45] in the worst case.

From Table 1, we can see that the proposed SRVR-HMC algorithm strictly outperforms HMC, UL-MCMC, SGHMC and SG-UL-MCMC, and also outperforms LMC, SGLD and SVRG-LD in terms of the dependency on target accuracy $\epsilon$ and training sample size $n$. We remark that for a general non-log-concave target density, $\lambda_*$ and $\mu_*$ are not directly comparable, though both of them are exponential in dimension $d$. However, it is shown that for a class of target densities, $\mu_*$ can be in the order of $O(\lambda_*^{1/2})$ [27, 30], which suggests that SRVR-HMC is also strictly better than LMC, SGLD and SVRG-LD for sampling from such densities.

**Notation.** We denote discrete update by lower case bold symbol $\mathbf{x}_k$ and continuous-time dynamics by upper case italicized bold symbol $\boldsymbol{X}_t$. For a vector $\mathbf{x} \in \mathbb{R}^d$, we denote by $\|\mathbf{x}\|_2$ the Euclidean norm. For random vectors $\mathbf{x}_k, \boldsymbol{X}_t \in \mathbb{R}^d$, we denote their probability distribution functions by $\mathbb{P}(\mathbf{x}_k)$ and $\mathbb{P}(\boldsymbol{X}_t)$ respectively. For a probability measure $\mu$, we denote by $\mathbb{E}_\mu[\boldsymbol{X}]$ the expectation of $\boldsymbol{X}$ under probability measure $u$. The 2-Wasserstein distance between two probability measures $u$ and $v$ is

$$\mathcal{W}_2(u,v) = \sqrt{\inf_{\zeta \in \Gamma(u,v)} \int_{\mathbb{R}^d \times \mathbb{R}^d} \|\boldsymbol{X}_u - \boldsymbol{X}_v\|_2^2 \mathrm{d}\zeta(\boldsymbol{X}_u, \boldsymbol{X}_v)},$$

where the infimum is taken over all joint distributions $\zeta$ with $u$ and $v$ being its marginal distributions. $\mathbb{1}(\cdot)$ denotes the indicator function. We denote index set $[n] = \{1, 2, \ldots, n\}$ for an integer $n$. We use $a_n = O(b_n)$ to denote that $a_n \leq Cb_n$ for some constant $C > 0$ independent of $n$, and use $a_n = \widetilde{O}(b_n)$ to hide the logarithmic factors in $b_n$. The Vinogradov notation $a_n \lesssim b_n$ is also used synonymously with $a_n = O(b_n)$. We denote $\min\{a, b\}$ and $\max\{a, b\}$ by $a \wedge b$ and $a \vee b$ respectively. The ceiling function $\lceil x \rceil$ outputs the least integer greater than or equal to $x$.

## 2 The proposed algorithm

In this section, we present our algorithm, SRVR-HMC, for sampling from a target distribution in the form of $\pi \propto \exp\{-f(\mathbf{x})\}$. Our algorithm is shown in Algorithm 1, which has a multi-epoch structure. In detail, there are $\lceil K/L \rceil$ epochs, where $K$ is the number of total iterations and $L$ denotes the epoch length, i.e., the number of iterations within each inner loop.

Recall that the update rule of HMC in (1.2) requires the computation of full gradient $\nabla f(\mathbf{x}_k)$ at each iteration, which is the average of $n$ stochastic gradients. This causes a high per-iteration complexity when $n$ is large. Therefore, we propose to leverage the stochastic gradient to offset the computational burden. At the beginning of the $j$-th epoch, we compute a stochastic gradient $\widetilde{\mathbf{g}}_j$ based on a batch of training data (uniformly sampled from $[n]$ without replacement) as shown in Line 4 of Algorithm 1, where the batch is denoted by $\widetilde{\mathcal{B}}_j$ with batch size $|\widetilde{\mathcal{B}}_j| = B_0$. In each epoch, we make use of the stochastic path-integrated differential estimator [29] to compute the following semi-stochastic gradient

$$\mathbf{g}_k = 1/B \sum_{i \in \mathcal{B}_k} \left[ \nabla f_i(\mathbf{x}_k) - \nabla f_i(\mathbf{x}_{k-1}) \right] + \mathbf{g}_{k-1}, \tag{2.1}$$

where $\mathcal{B}_k$ is another uniformly sampled (without replacement) mini-batch from $[n]$ with mini-batch size $|\mathcal{B}_k| = B$. Unlike the unbiased stochastic gradient estimators in SGHMC [16] and SVR-HMC [55], $\mathbf{g}_k$ is a biased estimator of the full gradient $\nabla f(\mathbf{x}_k)$ conditioned on $\mathbf{x}_k$. However, we can show that while being biased, the variance of $\mathbf{g}_k$ is substantially smaller than that of unbiased ones. This is the key reason why our algorithm can achieve a faster convergence rate than existing HMC-type algorithms. Based on the semi-stochastic gradient in (2.1), we update the position and velocity variables as follows

$$\begin{aligned}
\mathbf{v}_{k+1} &= \mathbf{v}_k e^{-\gamma\eta} - u\gamma^{-1}(1 - e^{-\gamma\eta})\mathbf{g}_k + \boldsymbol{\epsilon}_k^v, \\
\mathbf{x}_{k+1} &= \mathbf{x}_k + \gamma^{-1}(1 - e^{-\gamma\eta})\mathbf{v}_k + u\gamma^{-2}(\gamma\eta + e^{-\gamma\eta} - 1)\mathbf{g}_k + \boldsymbol{\epsilon}_k^x,
\end{aligned} \tag{2.2}$$

where $\eta$ is the step size and $u, \gamma$ are the inverse mass and friction parameter defined in (1.1), which are usually treated as tunable hyper parameters in practice. Moreover, $\boldsymbol{\epsilon}_k^v, \boldsymbol{\epsilon}_k^x \in \mathbb{R}^d$ are zero mean

---

**Algorithm 1** Stochastic Recursive Variance-Reduced gradient HMC (SRVR-HMC)

---

1: **input:** Initial points $\widetilde{\mathbf{x}}_0 = \mathbf{x}_0 = \mathbf{x}_0, \mathbf{v}_0$; step size $\eta$; batch sizes $B_0$ and $B$; total number of iterations $K$; epoch length $L$
2: **for** $j = 0, \ldots, \lceil K/L \rceil$ **do**
3:     Uniformly sample a subset of index $\widetilde{\mathcal{B}}_j \subset [n]$ with $|\widetilde{\mathcal{B}}_j| = B_0$
4:     Compute $\widetilde{\mathbf{g}}_j = 1/B_0 \sum_{i \in \widetilde{\mathcal{B}}_j} \nabla f_i(\widetilde{\mathbf{x}}_j)$
5:     **for** $l = 0, \ldots, L-1$ **do**
6:        $k = jL + l$
7:        **if** $l = 0$ **then**
8:           $\mathbf{g}_k = \widetilde{\mathbf{g}}_j$
9:        **else**
10:         Uniformly sample a subset of index $\mathcal{B}_k \subset [n]$ with $|\mathcal{B}_k| = B$
11:         Compute $\mathbf{g}_k = 1/B \sum_{i \in \mathcal{B}_k} (\nabla f_i(\mathbf{x}_k) - \nabla f_i(\mathbf{x}_{k-1})) + \mathbf{g}_{k-1}$
12:        **end if**
13:        $\mathbf{x}_{k+1} = \mathbf{x}_k + \gamma(1 - e^{-\gamma\eta})\mathbf{v}_k + u\gamma^{-2}(\gamma\eta + e^{-\gamma\eta} - 1)\mathbf{g}_k + \boldsymbol{\epsilon}_k^x$
14:        $\mathbf{v}_{k+1} = \mathbf{v}_k e^{-\gamma\eta} - u\gamma^{-1}(1 - e^{-\gamma\eta})\mathbf{g}_k + \boldsymbol{\epsilon}_k^v$
15:     **end for**
16:     $\widetilde{\mathbf{x}}_{j+1} = \mathbf{x}_{(j+1)L}$
17: **end for**
18: **output:** $\mathbf{x}_K$

---

Gaussian random vectors with covariance matrices satisfying

$$
\begin{aligned}
\mathbb{E}[\boldsymbol{\epsilon}_k^v(\boldsymbol{\epsilon}_k^v)^\top] &= u(1 - e^{-2\gamma\eta}) \cdot \mathbf{I}, \\
\mathbb{E}[\boldsymbol{\epsilon}_k^x(\boldsymbol{\epsilon}_k^x)^\top] &= u\gamma^{-2}(2\gamma\eta + 4e^{-\gamma\eta} - e^{-2\gamma\eta} - 3) \cdot \mathbf{I}, \\
\mathbb{E}[\boldsymbol{\epsilon}_k^v(\boldsymbol{\epsilon}_k^x)^\top] &= u\gamma^{-1}(1 - 2e^{-\gamma\eta} + e^{-2\gamma\eta}) \cdot \mathbf{I},
\end{aligned}
\tag{2.3}
$$

where $\mathbf{I} \in \mathbb{R}^{d \times d}$ is the identity matrix. The covariance of the Gaussian noises in (2.3) is obtained by integrating the Hamiltonian Langevin dynamics (1.1) over a time period of length $\eta$. It is worth noting our update rule in (2.2) and the construction of the Gaussian noises in (2.3) follow Cheng et al. [18], Zou et al. [55], Cheng et al. [17], except that we use a different semi-stochastic gradient estimator as shown in (2.1). In contrast, Cheng et al. [18] uses full gradient and noisy gradient, and Zou et al. [55] uses an unbiased semi-stochastic gradient based on SVRG [33].

We remark here that the semi-stochastic gradient estimator in (2.1) was originally proposed in finding stationary points in finite-sum optimization [44, 29] and further extended in [49, 32]. In addition, another semi-stochastic gradient estimator called SNVRG [54, 53] has also been demonstrated to achieve similar convergence rate in finite-sum optimization. Despite using the same semi-stochastic gradient estimator, our work differs from [44, 29] in at least two aspects: (1) the sampling problem studied in this paper is different from the optimization problem studied in [44, 29], where our goal is to sample from a target distribution concentrating on the global minimizer of $f(\mathbf{x})$ such that the sample distribution is close to the target distribution in 2-Wasserstein distance. In contrast, Nguyen et al. [44], Fang et al. [29] aim at finding a stationary point of $f(\mathbf{x})$ with small gradient; and (2) the algorithms in [44, 29] only have one update variable, while our SRVR-HMC algorithm has an additional Hamiltonian momentum term and therefore has two update variables (i.e., velocity and position variables). The Hamiltonian momentum is essential for underdamped Langevin Monte Carlo methods to achieve a smaller discretization error than overdamped methods such as SGLD [50] and SVRG-LD [25]. At the same time, this also introduces a great technical challenge in our theoretical analysis and requires nontrivial efforts.

## 3 Main theory

In this section, we provide the convergence guarantee for Algorithm 1. In particular, we characterize the 2-Wasserstein distance between the distribution of the output of Algorithm 1 and the target distribution $\pi \propto e^{-f(\mathbf{x})}$. We focus on sampling from non-log-concave densities that satisfy the smoothness and dissipativeness conditions, which are formally defined as follows.

**Assumption 3.1** (Smoothness). Each $f_i$ in (1.3) is $M$-smooth, i.e., there exists a positive constant $M > 0$, such that the following holds

$$\|\nabla f_i(\mathbf{x}) - \nabla f_i(\mathbf{y})\|_2 \leq M\|\mathbf{x} - \mathbf{y}\|_2, \quad \text{for any } \mathbf{x}, \mathbf{y} \in \mathbb{R}^d.$$

Note that Assumption 3.1 directly implies that function $f(\mathbf{x})$ is also $M$-smooth.

**Assumption 3.2** (Dissipativeness). There exist constants $m, b > 0$, such that the following holds

$$\langle \nabla f(\mathbf{x}), \mathbf{x} \rangle \geq m\|\mathbf{x}\|_2^2 - b, \quad \text{for any } \mathbf{x} \in \mathbb{R}^d.$$

Different from the smoothness assumption, Assumption 3.2 is only required for $f(\mathbf{x})$ rather than $f_i(\mathbf{x})$. The dissipativeness assumption is standard in the analysis for sampling from non-log-concave densities and is essential to guarantee the convergence of underdamped Langevin dynamics [46, 41].

### 3.1 Convergence analysis of the proposed algorithm

Now we state our main theorem that establishes the convergence rate of Algorithm 1.

**Theorem 3.3.** Suppose Assumptions 3.1 and 3.2 hold and the initial points are $\mathbf{x}_0 = \mathbf{v}_0 = \mathbf{0}$. If set $\gamma \leq 2\sqrt{Mu}$ and the step size $\eta \leq O(mM^{-3} \wedge m^{1/2}M^{-3/2}L^{-1/2})$, the output $\mathbf{x}_K$ of Algorithm 1 satisfies

$$\mathcal{W}_2\big(\mathbb{P}(\mathbf{x}_K), \pi\big) \leq \Gamma_1\bigg(\bigg(1 + \frac{L}{B}\bigg)K\eta^3 + \frac{K\eta}{\gamma^2 B_0} \cdot \mathbb{1}(B_0 < n)\bigg)^{1/4} + \Gamma_0 e^{-\mu_* K\eta},$$

where $B_0, B$ are the batch and minibatch sizes, $L$ is the epoch length and $\mu_* = \exp(-\widetilde{O}(d))$ is a lower bound of the spectral gap of the Markov process generated by (1.1). $\Gamma_0 = \widetilde{O}(\mu_*^{-1})$ and $\Gamma_1 = 2D_1(M^2\gamma^3 uD_2)^{1/4}$ are problem-dependent parameters with constants $D_1, D_2$ defined as

$$D_1 = \frac{8}{\gamma}\sqrt{\frac{um(f(\mathbf{0}) - f(\mathbf{x}^*)) + 2Mu(4d + 2b + m\|\mathbf{x}^*\|_2^2\gamma^2) + (12um + 3\gamma^2)}{m}},$$

$$D_2 = \frac{8um(f(\mathbf{0}) - f(\mathbf{x}^*)) + 8Mu\big(20(d + b) + m\|\mathbf{x}^*\|_2^2\big)}{\gamma^2 m} + \max_{i \in [n]} \frac{\|\nabla f_i(\mathbf{0})\|_2^2}{M^2},$$

and $\mathbf{x}^* = \text{argmin}_{\mathbf{x} \in \mathbb{R}^d} f(\mathbf{x})$ is the global minimizer of $f$.

Theorem 3.3 states that the 2-Wasserstein distance between the output of SRVR-HMC and the target distribution is upper bounded by two terms: the first term is the discretization error between the discrete-time Algorithm 1 and the continuous-time dynamics (1.1), which goes to zero when the step size $\eta$ goes to zero; the second term represents the ergodicity of the Markov process generated by (1.1) which converges to zero exponentially fast.

**Remark 3.4.** The result in Theorem 3.3 encloses a term $\mu_*$ with an exponential dependence on the dimension $d$, which is a lower bound of the spectral of the Markov process generated by (1.1). When $f$ is nonconvex, the exponential dependence of $\mu_*$ on dimension is unavoidable under the dissipativeness assumption [9]. However, this exponential dependency on $d$ can be weakened by imposing stronger assumptions on $f(\mathbf{x})$. For instance, Eberle et al. [27], Gao et al. [30] showed that for a symmetric double-well potential $f(\mathbf{x})$, $\mu_*$ is in the order of $\Omega(1/a)$, where $a$ is the distance between these two wells, and is typically polynomial in the dimension $d$. Another example is shown by Cheng et al. [17]: when $f(\mathbf{x})$ is strongly convex outside a $\ell_2$ ball centered at the origin with radius $R$, $\mu_*$ is in the order of $\exp(-O(MR^2))$ where $M$ is the smoothness parameter.

From Theorem 3.3, we can obtain the gradient complexity of SRVR-HMC by optimizing the choice of minibatch size $B$ and batch size $B_0$ in the following corollary.

**Corollary 3.5.** Under the same assumptions in Theorem 3.3, if set $B_0 = \widetilde{O}(\epsilon^{-4}\mu_*^{-1} \wedge n)$, $B \lesssim B_0^{1/2}$, $L = O(B_0/B)$, and $\eta = \widetilde{O}(\epsilon^2 B_0^{-1/2}\mu_*^{1/2}B)$, then Algorithm 1 requires $\widetilde{O}((n + \epsilon^{-2}n^{1/2}\mu_*^{-3/2}) \wedge \epsilon^{-4}\mu_*^{-2})$ stochastic gradient evaluations to achieve $\epsilon$-error in 2-Wasserstein distance.

**Remark 3.6.** Recall the gradient complexities of HMC and SGHMC in Table 1, it is evident that the gradient complexity of Algorithm 1 is lower than that of HMC [30] by a factor of $\widetilde{O}(\epsilon^{-2}n^{1/2}\mu_*^{3/2} \vee n\mu_*)$ and is lower than that of SGHMC [30] by a factor of $\widetilde{O}(\epsilon^{-6}n^{-1/2}\mu_*^{-7/2} \vee \epsilon^{-4}\mu_*^{-3})$.

**Remark 3.7.** As shown in Table 1, the gradient complexities of overdamped Langevin dynamics based algorithms, including LMC, SGLD and SVRG-LD, depend on the spectral gap $\lambda_*$ of the Markov chain generated by (1.4). Although the magnitudes of $\mu_*$ and $\lambda_*$ are not directly comparable, they are generally in the same order in the worst case [9, 45, 27]. Thus we treat them the same in the following comparison. In specific, the gradient complexity of SRVR-HMC is better than those of LMC [45] SGLD [45] and SVRG-LD [57] by factors of $\widetilde{O}(\epsilon^{-2}n^{1/2} \vee n)$, $\widetilde{O}(\epsilon^{-6}n^{-1/2} \vee \epsilon^{-4})$ and $\widetilde{O}(\epsilon^{-2} \vee n^{1/2})$ respectively.

## 3.2 Implication for UL-MCMC and SG-UL-MCMC

Recall the proposed SRVR-HMC algorithm in Algorithm 1, if we set the epoch length to be $L = 1$, Algorithm 1 degenerates to SG-UL-MCMC [18], with the following update formulation:

$$
\begin{aligned}
\mathbf{v}_{k+1} &= \mathbf{v}_k e^{-\gamma\eta} - u\gamma^{-1}(1 - e^{-\gamma\eta})\widetilde{\mathbf{g}}_k + \boldsymbol{\epsilon}_k^v, \\
\mathbf{x}_{k+1} &= \mathbf{x}_k + \gamma^{-1}(1 - e^{-\gamma\eta}\mathbf{v}_k) + u\gamma^{-2}(\gamma\eta + e^{-\gamma\eta} - 1)\widetilde{\mathbf{g}}_k + \boldsymbol{\epsilon}_k^x,
\end{aligned}
\tag{3.1}
$$

where $\widetilde{\mathbf{g}}_k = |\widetilde{\mathcal{B}}_k|^{-1} \sum_{i=1}^n \nabla f_i(\mathbf{x}_k)$ denotes the stochastic gradient computed in the $k$-th iteration. In addition, if we replace $\widetilde{\mathbf{g}}_k$ with the full gradient $\nabla f(\mathbf{x}_k)$, SG-UL-MCMC in (3.1) further reduces to UL-MCMC [18]. Although these two algorithms were originally proposed for sampling from strongly-log-concave densities [18], in this subsection, we show that our analysis of SRVR-HMC can be easily adapted to derive the gradient complexity of UL-MCMC/SG-UL-MCMC for sampling from non-log-concave densities. We first state the convergence of SG-UL-MCMC in the following theorem.

**Theorem 3.8.** Under the same assumptions in Theorem 3.3, the output $\mathbf{x}_K$ of the SG-UL-MCMC algorithm in (3.1) satisfies

$$
\mathcal{W}_2\big(\mathbb{P}(\mathbf{x}_K), \pi\big) \leq \Gamma_1\big[2K\eta^3 + K\eta/(\gamma^2 B_0) \cdot \mathbb{1}(B_0 < n)\big]^{1/4} + \Gamma_0 e^{-\mu_* K\eta},
$$

where $B_0$ denotes the mini-batch size, $\mu_*, \Gamma_0$ and $\Gamma_1$ are defined in Theorem 3.3.

Similar to the results in Theorem 3.3, the sampling error of SG-UL-MCMC in 2-Wasserstein distance is also controlled by the discretization error of the discrete algorithm (3.1) and the ergodicity rate of Hamiltonian Langevin dynamics (1.1). In particular, the main difference in the convergence results of SG-UL-MCMC and SRVR-HMC lies in the discretization error term, which leads to a different gradient complexity for SG-UL-MCMC.

**Corollary 3.9.** Under the same assumptions in Theorem 3.3, if we set $\eta = \widetilde{O}(\epsilon^2 \mu_*^{1/2})$ and $B_0 = \widetilde{O}(\epsilon^{-4}\mu_*^{-1})$, SG-UL-MCMC in (3.1) requires $\widetilde{O}(\epsilon^{-6}\mu_*^{-5/2})$ stochastic gradient evaluations to achieve $\epsilon$-error in 2-Wasserstein distance. Moreover, UL-MCMC requires $\widetilde{O}(\epsilon^{-2}\mu_*^{-3/2}n)$ stochastic gradient evaluations to achieve $\epsilon$-error in 2-Wasserstein distance.

**Remark 3.10.** Our theoretical analysis suggests that the gradient complexity of UL-MCMC is better than that of HMC [30] by a factor of $O(\epsilon^{-2}\mu_*^{-3/2})$ and the gradient complexity of SG-UL-MCMC is better than that of SGHMC [30] by a factor of $O(\epsilon^{-2}\mu_*^{-5/2})$. We note that Cheng et al. [17] proved $O(1/\epsilon)$ convergence rate of UL-MCMC for sampling from a smaller class of non-log-concave densities in 1-Wasserstein distance. Their result is not directly comparable to our result since 1-Wasserstein distance is strictly smaller than 2-Wasserstein distance and more importantly, their results rely on a stronger assumption than the dissipativeness assumption used in our paper as we commented in Remark 3.4.

# 4 Experiments

In this section, we evaluate the empirical performance of SRVR-HMC on both synthetic and real datasets. We compare our proposed algorithm with existing overdamped and underdamped Langevin based stochastic gradient algorithms including SGLD [50], SVRG-LD [25], SGHMC [16], SG-UL-MCMC [18] and SVR-HMC [55].

## 4.1 Sampling from Gaussian mixture distributions

We first demonstrate the performance of SRVR-HMC for fitting a Gaussian mixture model on synthetic data . In this case, the density on each data point is defined as

$$e^{-f_i(\mathbf{x})} = 2e^{\|\mathbf{x}-\mathbf{a}_i\|_2^2/2} + e^{\|\mathbf{x}+\mathbf{a}_i\|_2^2/2},$$

which is proportional to the probability density function (PDF) of two-component Gaussian mixture density with weights $1/3$ and $2/3$. By simple calculation, it can be verified that when $\|\mathbf{a}_i\|_2 \geq 1$, $f_i(\mathbf{x})$ is nonconvex but satisfies Assumption 3.2, and so does $f(\mathbf{x}) = 1/n \sum_{i=1}^{n} f_i(\mathbf{x})$.

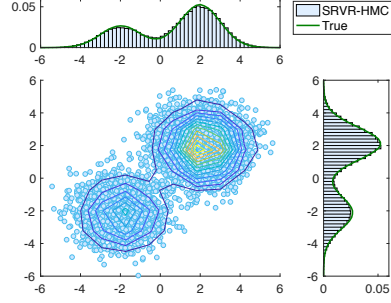

We generated $n = 500$ vectors $\{\mathbf{a}_i\}_{i=1,...,n} \in \mathbb{R}^2$ to construct the target density functions. We first show that the proposed algorithm can well approximate the target distribution. Specifically, we run SRVR-HMC for $10^4$ data passes, and use the last $10^5$ iterates to visualize the estimated distribution, where the batch size, minibatch size and epoch length are set to be $B_0 = n$, $B = 1$ and $L = n$ respectively. As a reference, we run MCMC with Metropolis-Hasting (MH) correction to represent the underlying distribution. Following [3], we display the kernel densities of random samples generated by SRVR-HMC in Figures 4.1, which shows that the random samples generated by SRVR-HMC well approximate Gaussian mixture distribution.

Figure 1: Kernel density estimation for Gaussian mixture distribution.

In Figure 2(a), we compare the performance of SRVR-HMC with baseline algorithms for sampling from Gaussian mixture distribution. Since directly computing the 2-Wasserstein distance is expensive, we resort to the mean square error (MSE) $\mathbb{E}[\|\hat{\mathbf{x}} - \bar{\mathbf{x}}\|_2^2]$, where $\bar{\mathbf{x}} = \mathbb{E}_\pi[\mathbf{x}]$ is obtained via running MCMC with MH correction and $\hat{\mathbf{x}} = \sum_{s=1001}^{k} \mathbf{x}_s/(k - 1000)$ is the sample path average, where $\mathbf{x}_s$ denotes the $s$-th position iterate of the algorithms and we discard the first 1000 iterates as burn-in. We report the MSE results of all algorithms in Figure 2(a) by repeating each algorithms for 20 times. It can be seen that SRVR-HMC converges faster than all baseline algorithms,

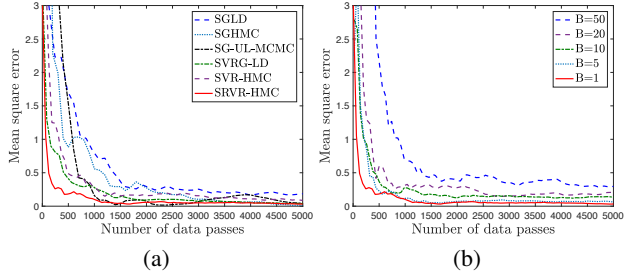

(a)                    (b)

Figure 2: Experiment results for sampling from Gaussian mixture distribution, where X-axis represents the number of data passes and Y-axis represents MSE: (a) Comparison with baseline algorithms. (b) Convergence of SRVR-HMC with varying batch size $B$.

which is well aligned with our theory. In addition, it can be seen SG-UL-MCMC outperforms SGHMC, which is consistent with our results in Table 1. We also compare the convergence performance of SRVR-HMC with different batch sizes in Figure 2(b). It can be observed that SRVR-HMC works well for all small batch sizes ($B < 20$) but becomes significantly worse when $B$ is large ($B = 50$). This observation is consistent with Corollary 3.5 where we prove that when $B \lesssim B_0^{1/2}$ the gradient complexity maintains the same.

## 4.2 Independent components analysis

We further run the sampling algorithms for independent components analysis (ICA) tasks. In the ICA model, the input are examples $\{\mathbf{x}_i\}_{i=1}^{n}$, and the likelihood function can be written as $p(\mathbf{x}|\mathbf{W}) = |\det(\mathbf{W})| \prod_{j=1}^{l} p(\mathbf{w}_j^\top \mathbf{x})$, where $\mathbf{W} \in \mathbb{R}^{d \times l}$ is the model matrix, $d$ is the problem dimension, $l$ denotes the number of independent components and $\mathbf{w}_j$ denotes the $j$-th column of $\mathbf{W}$. Following [50, 25] we set $p(\mathbf{w}_j^\top \mathbf{x}) = 1/(4\cosh^2(\mathbf{w}_j^\top \mathbf{x}/2))$ with a Gaussian prior $p(\mathbf{W}) \sim \mathcal{N}(0, \lambda^{-1}\mathbf{I})$. Then the negative log-posterior can be written as $f(\mathbf{W}) = 1/n \sum_{i=1}^{n} f_i(\mathbf{W})$, where

$$f_i(\mathbf{W}) = -n\log(|\det(\mathbf{W})|) - 2n\sum_{j=1}^{l}\log\left(\cosh(\mathbf{w}_j^\top \mathbf{x}_i/2)\right) + \lambda\|\mathbf{W}\|_F^2/2.$$

We compare the performance of SRVR-HMC with all the baseline algorithms on MEG dataset[4], which consists of 17730 time-points in 122 channels. In order to explore the performance of our

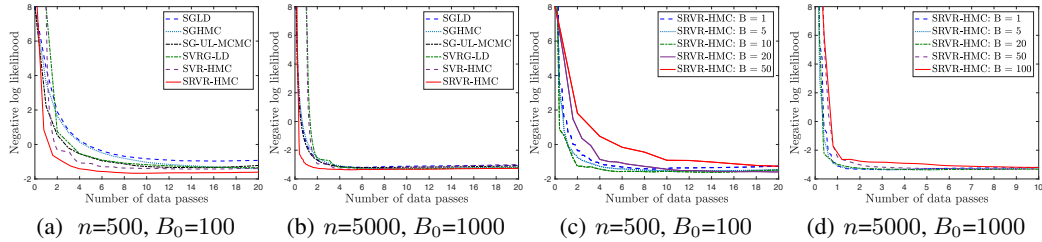

(a) $n$=500, $B_0$=100    (b) $n$=5000, $B_0$=1000    (c) $n$=500, $B_0$=100    (d) $n$=5000, $B_0$=1000

Figure 3: Experiment results for ICA, where X-axis represents the number of data passes, and Y-axis represents the negative log likelihood on the test dataset: (a)-(b) Comparison with different baselines (c)-(d) Convergence of SRVR-HMC with varying batch size $B$.

algorithm for different sample size, we extract two subset with sizes $n = 500$ and $n = 5000$ from the original dataset for training, and regard the rest $12730$ examples as test dataset. For inference, we compute the sample path average while discarding the first $100$ iterates as burn-in. We first compare the convergence performance of SRVR-HMC with baseline algorithms and report the negative log likelihood on test dataset in Figures 3(a)-3(b), where the batch size, minibatch size and epoch length are set to be $B_0 = n/5$, $B = 10$ and $L = B_0/B$, and the rest hyper parameters are tuned to achieve the best performance. It is worth noting that we do not perform the normalization when evaluating the test likelihood, thus the negative log likelihood results may be smaller than $0$. From Figures 3(a)-3(b) it can be clearly seen that SRVR-HMC outperforms all baseline algorithms, which validates its superior theoretical properties. Again, we can see that SG-UL-MCMC can decrease the negative log likelihood much faster than SGHMC, which is well aligned with our theory. Furthermore, we evaluate the convergence for different minibatch size, which are displayed in Figures 3(c)-3(d), where the batch size $B_0$ is fixed as $n/5$ for both scenarios. It can be seen that SRVR-HMC attains similar convergence performance for all small minibatch sizes ($B \leq 10$ when $B_0 = 100$ and $B \leq 20$ when $B_0 = 1000$), which again corroborates our theory that when $B \lesssim B_0^{1/2}$ the gradient complexity maintains the same.

We also evaluate our proposed algorithm SRVR-HMC on Bayesian logistic regression. We defer the additional experimental results to Appendix E due to space limit.

## 5   Conclusions

We propose a novel algorithm SRVR-HMC based on Hamiltonian Langevin dynamics for sampling from a class of non-log-concave target densities. We show that SRVR-HMC achieves a lower gradient complexity in 2-Wasserstein distance than all existing HMC-type algorithms. In addition, we show that our algorithm reduces to UL-MCMC and SG-UL-MCMC with properly chosen parameters. Our analysis of SRVR-HMC directly applies to these two algorithms and suggests that UL-MCMC/SG-UL-MCMC are faster than HMC/SGHMC for sampling from non-log-concave densities.

## Acknowledgement

We would like to thank the anonymous reviewers for their helpful comments. This research was sponsored in part by the National Science Foundation BIGDATA IIS-1855099 and CAREER Award IIS-1906169. The views and conclusions contained in this paper are those of the authors and should not be interpreted as representing any funding agencies.

## Footnotes

[1]$\widetilde{O}(\cdot)$ hides constant and logarithm factors.

[2]Gradient complexity is the total number of stochastic gradients $\nabla f_i(\mathbf{x})$ an algorithm needs to compute in order to achieve $\epsilon$-error in terms of certain measurement.

[3]The original results for LMC/SGLD in [45] and for HMC/SG-HMC in [30] are about the global convergence in nonconvex optimization. Yet their results can be adapted to sampling from non-log-concave distributions, and the corresponding gradient complexities can be spelled out from their convergence rates.

[4]http://research.ics.aalto.fi/ica/eegmeg/MEG_data.html

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
