[Supplementary Material]

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

# A  Proof of main theory

In this section, we prove our main theorems and corollaries.

## A.1  Proof of Theorem 3.3

We first prove the convergence of Algorithm 1 in terms of 2-Wasserstein distance. To simplify the proof, we define the following concatenated vectors

$$\mathbf{z}_k = \begin{pmatrix} \mathbf{x}_k \\ \mathbf{v}_k \end{pmatrix}, \qquad \mathbf{Z}_t = \begin{pmatrix} \mathbf{X}_t \\ \mathbf{V}_t \end{pmatrix} \tag{A.1}$$

where $\mathbf{x}_k, \mathbf{v}_k$ are the iterates in Algorithm 1 and $\mathbf{X}_t, \mathbf{V}_t$ are the variables in the continuous-time dynamics (1.1). Instead of directly bounding $\mathcal{W}_2(\mathbb{P}(\mathbf{x}_k), \pi)$, we aim to prove that its upper bound $\mathcal{W}_2(\mathbb{P}(\mathbf{z}_k), \pi_{\mathbf{z}})$ converges to $\epsilon$-precision, where

$$\pi_{\mathbf{z}} \propto \exp(-\|\mathbf{v}\|_2^2/(2u) + f(\mathbf{x})) \tag{A.2}$$

denotes the stationary distribution of Hamiltonian dynamics (1.1) with respect to both $\mathbf{x}$ and $\mathbf{v}$. By triangle inequality, it holds that

$$\mathcal{W}_2\big(\mathbb{P}(\mathbf{z}_k), \pi_{\mathbf{z}}\big) \leq \mathcal{W}_2\big(\mathbb{P}(\mathbf{z}_k), \mathbb{P}(\mathbf{Z}_{k\eta})\big) + \mathcal{W}_2\big(\mathbb{P}(\mathbf{Z}_{k\eta}), \pi_{\mathbf{z}}\big). \tag{A.3}$$

The first term on the R.H.S. of (A.3) represents the discretization error of Algorithm 1, and the second term is typically referred to the ergodicity of the continuous-time dynamics (1.1), which characterizes the mixing time of the Markov process $(\mathbf{X}_t, \mathbf{V}_t)$. These two terms can be upper bounded by the following lemmas respectively.

**Lemma A.1.** Suppose the initial point of Algorithm 1 is $\mathbf{x} = \mathbf{v} = \mathbf{0}$. $\mathbf{z}_k$ and $\mathbf{Z}_t$ are defined as in (A.1). Under Assumptions 3.1 and 3.2, if we set the step size $\eta = O(mM^{-3} \wedge m^{1/2}M^{-3/2}L^{-1/2})$, the 2-Wasserstein distance between the iterate $\mathbf{z}_k$ generated by Algorithm 1 and the point $\mathbf{Z}_{k\eta}$ generated by Hamiltonian dynamics (1.1) is upper bounded as follows,

$$\mathcal{W}_2\big(\mathbb{P}(\mathbf{z}_k), \mathbb{P}(\mathbf{Z}_{k\eta})\big) \leq 2\bar{\Lambda}\bigg( M^2\gamma^3 u\bar{\mathcal{E}}\Big(1 + \frac{L}{B}\Big)K\eta^3 + \frac{M^2\gamma u\bar{\mathcal{E}}K\eta}{B_0} \cdot \mathbb{1}(B_0 < n) \bigg)^{1/4},$$

where $\bar{\Lambda}$ and $\bar{\mathcal{E}}$ are defined as

$$\bar{\Lambda} = \frac{8}{\gamma}\sqrt{\frac{um(f(\mathbf{x}_0) - f(\mathbf{x}^*)) + 2Mu(4d + 2b + m\|\mathbf{x}^*\|_2^2\gamma^2) + (12um + 3\gamma^2)}{m}},$$

$$\bar{\mathcal{E}} = \frac{8um(f(\mathbf{x}_0) - f(\mathbf{x}^*)) + 8Mu\big(20(d + b) + m\|\mathbf{x}^*\|_2^2\big)}{\gamma^2 m} + \frac{G^2}{M^2},$$

$G = \max_{i \in n}\|\nabla f_i(0)\|_2$ and $\mathbf{x}^* = \operatorname{argmin}_{\mathbf{x}\mathbb{R}^d} f(\mathbf{x})$ is the global minimizer of $f$.

**Lemma A.2.** $\mathbf{Z}_t$ and $\pi_{\mathbf{z}}$ are defined as in (A.1) and (A.2) respectively. Under Assumptions 3.1 and 3.2, then we have

$$\mathcal{W}_2\big(\mathbb{P}(\mathbf{Z}_t), \pi_{\mathbf{z}}\big) \leq \Gamma_0 e^{-\mu_* t},$$

where $\mu_*$ denotes the contraction rate of Hamiltonian Langevin dynamics (1.1), which is in the order of $e^{-\widetilde{O}(d)}$ under Assumption 3.2, and $\Gamma_0$ is a constant of order $O(1/\mu_*)$.

Here $\mu^*$ serves as a lower bound of the spectral gap of the spectral gap of the Markov process generated by (1.1), and in the worst case the exponential dependency on $d$ is unavoidable [27].

Based on the above two lemmas, the proof of Theorem 3.3 is straightforward.

*Proof of Theorem 3.3.* By Lemmas A.1 and A.2, it holds that

$$\mathcal{W}_2\big(\mathbb{P}(\mathbf{z}_K), \pi_{\mathbf{z}}\big) \leq \mathcal{W}_2\big(\mathbb{P}(\mathbf{z}_K), \mathbb{P}(\mathbf{Z}_{K\eta})\big) + \mathcal{W}_2\big(\mathbb{P}(\mathbf{Z}_{K\eta}), \pi_{\mathbf{z}}\big)$$

$$\leq \Gamma_1\bigg[\Big(1 + \frac{L}{B}\Big)K\eta^3 + \frac{K\eta}{\gamma^2 B_0} \cdot \mathbb{1}(B_0 < n)\bigg]^{1/4} + \Gamma_0 e^{-\mu_* K\eta},$$

where $\Gamma_1 = 2D_1(M^2\gamma^3 uD_2)^{1/4}$, $\Gamma_0$ is defined in A.2, and $D_1$, $D_2$ correspond to $\bar{\Lambda}$ and $\bar{\mathcal{E}}$ in Lemmas A.1 respectively. By plugging in the definition of 2-Wasserstein distance, we obtain the fact that $\mathcal{W}_2\big(\mathbb{P}(\mathbf{x}_K), \pi\big) \leq \mathcal{W}_2\big(\mathbb{P}(\mathbf{z}_K), \pi_{\mathbf{z}}\big)$, which completes the proof. $\qquad\square$

## A.2 Proof of Corollary 3.5

*Proof.* In order to ensure the 2-Wasserstein distance $\mathcal{W}_2\big(\mathbb{P}(\mathbf{x}_k), \pi\big) \leq \epsilon$, we can set

$$\Gamma_1 \left[ \left(1 + \frac{L}{B}\right) K\eta^3 + \frac{K\eta}{\gamma^2 B_0} \cdot \mathbb{1}(B_0 < n) \right]^{1/4} \leq \frac{\epsilon}{2} \quad \text{and} \quad \Gamma_0 e^{-\mu_* K\eta} = \frac{\epsilon}{2}. \qquad \text{(A.4)}$$

For the first equation in (A.4), we further set $\eta$ sufficiently small and $B_0$ sufficiently large such that

$$\left(1 + \frac{L}{B}\right) K\eta^3 = \frac{1}{2}\left(\frac{\epsilon}{2\Gamma_1}\right)^4 \quad \text{and} \quad \frac{K\eta}{\gamma^2 B_0} \cdot \mathbb{1}(B_0 < n) \leq \frac{1}{2}\left(\frac{\epsilon}{2\Gamma_1}\right)^4.$$

Solving the second equation in (A.4), we obtain $K\eta = \mu_*^{-1}\log(2\Gamma_0/\epsilon)$. Plugging this into the above equations, we have

$$\eta = \frac{\epsilon^2 \mu_*^{1/2}}{4\sqrt{2}\Gamma_1^2 \sqrt{(1 + L/B)\log(2\Gamma_0/\epsilon)}}, \qquad B_0 = \frac{32\Gamma_1^4 \log(2\Gamma_0/\epsilon)}{\epsilon^4 \gamma^2 \mu_*} \wedge n.$$

Combining the choice of $\eta$ and the fact that $K\eta = \mu_*^{-1}\log(2\Gamma_0/\epsilon)$, we get

$$K = \frac{4\sqrt{2}\Gamma_1^2 (1 + L/B)^{1/2} \log^{3/2}(2\Gamma_0/\epsilon)}{\epsilon^2 \mu_*^{3/2}}.$$

Now we can calculate the total gradient complexity of Algorithm 1 as follows:

$$T_g = KB + KB_0/L + B_0.$$

In order to minimize the gradient complexity, it requires to set $BL = O(B_0)$ which implies that $\eta = O(\epsilon^2 \mu_*^{1/2} B_0^{-1/2} B)$. Then we have

$$T_g = \widetilde{O}\big(B_0 + (B^2 + LB)^{1/2} \mu_*^{-3/2} \epsilon^{-2}\big).$$

Note that $B_0 = \widetilde{O}(\epsilon^{-4} \wedge n)$. Thus we can chose the set size $B$ such that $B^2 \lesssim B_0$ and get

$$T_g = \widetilde{O}(\epsilon^{-2}\mu_*^{-3/2}B_0^{1/2}) + O(B_0) = \widetilde{O}\big(\big(\epsilon^{-2}\mu_*^{-3/2}n^{1/2} + n\big) \wedge \epsilon^{-4}\mu_*^{-2}\big).$$

This completes the proof. □

## A.3 Proof of Theorem 3.8

*Proof of Theorem 3.8.* As we mentioned before, the proposed algorithm can reduce to a variant of SGHMC algorithm by setting $L = 1$. Therefore, the convergence guarantee of SGHMC can be directly generalized from Theorem 3.3, i.e.,

$$\mathcal{W}_2\big(\mathbb{P}(\mathbf{x}_K), \pi\big) \leq \Gamma_1 \left[ \left(1 + \frac{1}{B}\right) K\eta^3 + \frac{K\eta}{\gamma^2 B_0} \cdot \mathbb{1}(B_0 < n) \right]^{1/4} + \Gamma_0 e^{-\mu_* K\eta}.$$

Since $B \geq 1$, we have $1/B \leq 1$. Plugging this into the above inequality, we can complete the proof. □

## A.4 Proof of Corollary 3.9

*Proof of Corollary 3.9.* In order to guarantee the distance $\mathcal{W}_2\big(\mathbb{P}(\mathbf{x}_K), \pi\big)$ be smaller than $\epsilon$, we can set

$$2K\eta^3 = \frac{\epsilon^4}{32\Gamma_1^4}, \quad \frac{K\eta}{\gamma^2 B_0}\mathbb{1}(B_0 < n) = \frac{\epsilon^4}{32\Gamma_1^4} \quad \text{and} \quad \Gamma_0 e^{-\mu_* K\eta} = \frac{\epsilon}{2}.$$

Solving the above equations, we get

$$K\eta = \mu_*^{-1}\log(2\Gamma_0/\epsilon), \quad \eta = \frac{\epsilon^2 \mu_*^{1/2}}{8\Gamma_1^2 \sqrt{\log(2\Gamma_0/\epsilon)}}, \quad B_0 = \frac{32\Gamma_1^4 \log(2\Gamma_0/\epsilon)}{\epsilon^4 \gamma^2 \mu_*} \wedge n.$$

Solving the above we further obtain

$$K = \frac{8\Gamma_1^2 \log^{3/2}(2\Gamma_0/\epsilon)}{\epsilon^2 \mu_*^{3/2}},$$

which implies that the gradient complexity of SG-UL-MCMC is

$$T = KB_0 = \widetilde{O}\big(\epsilon^{-6}\mu^{-5/2} \wedge \epsilon^{-2}\mu^{-3/2}n\big).$$

This completes the proof. □

# B  Proof of technical lemmas

In this section, we provide the proofs of the two key lemmas presented in the analysis in Appendix A.

## B.1  Proof of Lemma A.1

We first lay down the supporting lemmas that would be useful in our proof.

**Lemma B.1** (Lemma 10 in [18]). *The Hamiltonian Langevin dynamics (1.1) has the following solution*

$$\boldsymbol{V}_t = \boldsymbol{V}_0 e^{-\gamma t} - u \int_0^t e^{-\gamma(t-s)} \nabla f(\boldsymbol{X}_t) \mathrm{d}s + \widetilde{\boldsymbol{\epsilon}}_t^v, \tag{B.1}$$

$$\boldsymbol{X}_t = \boldsymbol{X}_0 + \frac{1-e^{-\gamma t}}{\gamma} \boldsymbol{V}_0 + u \int_0^t \int_0^s e^{-\gamma(s-r)} \nabla f(\boldsymbol{X}_r) \mathrm{d}r \mathrm{d}s + \widetilde{\boldsymbol{\epsilon}}_t^x, \tag{B.2}$$

*where $\widetilde{\boldsymbol{\epsilon}}_t^v = \sqrt{2\gamma u} \int_0^t e^{-\gamma(t-s)} \mathrm{d}\boldsymbol{B}_s$ and $\widetilde{\boldsymbol{\epsilon}}_t^x = \sqrt{2\gamma u} \int_0^t \int_0^s e^{-\gamma(s-r)} \mathrm{d}\boldsymbol{B}_r \mathrm{d}s$ are Gaussian random variables with mean $\mathbf{0}$ and their covariance matrices are as follows:*

$$\mathbb{E}[\widetilde{\boldsymbol{\epsilon}}_t^v (\widetilde{\boldsymbol{\epsilon}}_t^v)^\top] = u(1 - e^{-2\gamma t}) \cdot \mathbf{I}_{d \times d}$$

$$\mathbb{E}[\widetilde{\boldsymbol{\epsilon}}_t^x (\widetilde{\boldsymbol{\epsilon}}_t^x)^\top] = \frac{u}{\gamma^2}(2\gamma t + 4e^{-\gamma t} - e^{-2\gamma t} - 3) \cdot \mathbf{I}_{d \times d}$$

$$\mathbb{E}[\widetilde{\boldsymbol{\epsilon}}_t^v (\widetilde{\boldsymbol{\epsilon}}_t^x)^\top] = \frac{u}{\gamma}(1 - 2e^{-\gamma t} + e^{-2\gamma t}) \cdot \mathbf{I}_{d \times d}.$$

To prove the convergence of Algorithm 1, we define a Lyapunov function for all $(\mathbf{x}, \mathbf{v}) \in \mathbb{R}^d \times \mathbb{R}^d$ as follows

$$\mathcal{E}(\mathbf{x}, \mathbf{y}) = \|\mathbf{x}\|_2^2 + \|\mathbf{x} + 2\mathbf{v}/\gamma\|_2^2 + 8u(f(\mathbf{x}) - f(\mathbf{x}^*))/\gamma^2. \tag{B.3}$$

Note that $\|\mathbf{a}\|_2^2 + \|\mathbf{b}\|_2^2 \geq \|\mathbf{a} - \mathbf{b}\|_2^2/2$. By the definition of $\mathcal{E}$ and the fact that $f(\mathbf{x}) \geq f(\mathbf{x}^*)$, we have

$$\mathcal{E}(\mathbf{x}, \mathbf{v}) \geq \|\mathbf{x}\|_2^2 + \|\mathbf{x} + 2\mathbf{v}/\gamma\|_2^2 \geq \max\{\|\mathbf{x}\|_2^2, 2\|\mathbf{v}/\gamma\|_2^2\}. \tag{B.4}$$

**Lemma B.2.** *Under Assumptions 3.1 and 3.2, if we set the step size of Algorithm 1 according to the following condition:*

$$\eta \leq \min\left( \frac{\gamma}{4(8Mu + u\gamma + 22\gamma^2)}, \sqrt{\frac{4u^2}{4Mu + 3\gamma^2}}, \frac{6\gamma bu}{(4Mu + 3\gamma^2)d}, \right.$$

$$\left. \frac{\gamma^4 m}{48(46\gamma^2 + 288u\gamma + 32u)M^3 u}, \frac{\gamma m^{1/2}}{48M^{3/2}(\gamma^2 + u)^{1/2}L^{1/2}}, \frac{\gamma}{\sqrt{6Mu}}, \frac{\gamma \bar{\mathcal{E}}^{1/2}}{2Gu}, \frac{1}{2\sqrt{L}\gamma} \right),$$

*and $B_0 \geq \min\{1/\eta, n\}$, then for all $k \geq 0$, $\mathbb{E}[\|\mathbf{x}_k\|_2^2]$, $\mathbb{E}[\|\mathbf{v}_k\|_2^2]$ and $\mathbb{E}[\|\mathbf{g}_k\|_2^2]$ can be bounded as follows,*

$$\mathbb{E}[\|\mathbf{x}_k\|_2^2] \leq \bar{\mathcal{E}}, \quad \mathbb{E}[\|\mathbf{v}_k\|_2^2] \leq \gamma^2 \bar{\mathcal{E}}/2, \quad \text{and} \quad \mathbb{E}[\|\mathbf{g}_k\|_2^2] \leq 14M^2 \bar{\mathcal{E}},$$

*where $\bar{\mathcal{E}}$ is defined as*

$$\bar{\mathcal{E}} = \mathcal{E}(\mathbf{x}_0, \mathbf{v}_0) + \frac{8Mu[16(d+b) + m\|\mathbf{x}^*\|_2^2]}{\gamma^2 m} + \frac{G^2}{M^2}, \qquad G = \max_{i \in n} \|\nabla f_i(\mathbf{0})\|_2,$$

*and $\mathcal{E}(\mathbf{x}, \mathbf{y})$ is the Lyapunov function defined in (B.3).*

The following lemma characterizes the expected distance between the semi-stochastic gradient $\mathbf{g}_k$ and the full gradient $\nabla f(\mathbf{x}_k)$.

**Lemma B.3.** *Suppose Assumptions 3.1 and 3.2 hold. For Algorithm 1, if we choose the same step size $\eta$ used in Lemma B.2, then it holds that*

$$\mathbb{E}[\|\mathbf{g}_k - \nabla f(\mathbf{x}_k)\|_2^2] \leq \frac{4LM^2\gamma^2\eta^2\bar{\mathcal{E}}}{B} + \frac{4M^2\bar{\mathcal{E}}}{B_0} \cdot \mathbb{1}(B_0 < n),$$

*where $\bar{\mathcal{E}}$ is defined in Lemma B.2.*

The next lemma is referred to as the exponential integrability.

**Lemma B.4.** Suppose Assumptions 3.1 and 3.2 hold. Let $\theta > 0$ be any constant such that $\theta \leq \min\{\gamma^2/(128u), m/32\}$. Then, it holds that

$$\log\left(\mathbb{E}\big[e^{\theta(\|\boldsymbol{X}_t\|_2^2 + \|\boldsymbol{V}_t\|_2^2)}\big]\right) \leq 2\theta\mathcal{E}(\boldsymbol{X}_0, \boldsymbol{V}_0) + \frac{32M\theta u\big(4d + 2b + m\|\mathbf{x}^*\|_2^2\big)}{\gamma^2 m},$$

where $\mathcal{E}(\mathbf{x}, \mathbf{y})$ is the Lyapunov function defined in (B.3).

The following weighted CKP inequality gives a tight connection between 2-Wasserstein distance and KL divergence.

**Lemma B.5** (Weighted CKP Inequality [7]). For any two probability measures $P$ and $Q$, if they have finite second moments, the following holds,

$$\mathcal{W}_2(Q, P) \leq \Lambda(\sqrt{D_{KL}(Q\|P)} + \sqrt[4]{D_{KL}(Q\|P)}),$$

where $\Lambda = 2\inf_{\theta>0}\sqrt{1/\theta(3/2 + \log\mathbb{E}_{\mathbf{x}\sim P}[e^{\theta\|\mathbf{x}\|_2^2}])}$.

Now we are ready to prove our first key lemma on the discretization error of Algorithm 1.

*Proof of Lemma A.1.* By the weighted CKP inequality in Lemma B.5, we have

$$\mathcal{W}_2\big(\mathbb{P}(\mathbf{z}_K), \mathbb{P}(\boldsymbol{Z}_{K\eta})\big) \leq \Lambda\Big(\sqrt{D_{KL}(\mathbb{P}(\mathbf{z}_K)\|\mathbb{P}(\boldsymbol{Z}_{K\eta}))} + \sqrt[4]{D_{KL}(\mathbb{P}(\mathbf{z}_K)\|\mathbb{P}(\boldsymbol{Z}_{K\eta}))}\Big), \quad \text{(B.5)}$$

where $\Lambda = 2\inf_{\theta>0}\sqrt{1/\theta(3/2 + \log\mathbb{E}_{\mathbb{P}(\boldsymbol{Z}_T)}[e^{\theta\|\boldsymbol{Z}_T\|_2^2}])}$ and $T = K\eta$. By (A.1) it holds that $\|\boldsymbol{Z}_T\|_2^2 = \|\boldsymbol{X}_T\|_2^2 + \|\boldsymbol{V}_T\|_2^2$. Applying Lemma B.4, we obtain

$$\Lambda = 2\inf_{\theta>0}\sqrt{1/\theta(3/2 + \log\mathbb{E}_{\mathbb{P}_T}[e^{\theta(\|\boldsymbol{X}_T\|_2^2 + \|\boldsymbol{V}_T\|_2^2)}])}$$

$$\leq 2\inf_{0<\theta\leq\min\{\frac{\gamma^2}{128u}, \frac{m}{32}\}}\sqrt{\frac{1}{\theta}\left(\frac{3}{2} + 2\theta\mathcal{E}(\boldsymbol{X}_0, \boldsymbol{V}_0) + \frac{32M\theta u\big(4d + 2b + m\|\mathbf{x}^*\|_2^2\big)}{\gamma^2 m}\right)}$$

$$\leq 2\sqrt{2\mathcal{E}(\boldsymbol{X}_0, \boldsymbol{V}_0) + \frac{32Mu\big(4d + 2b + m\|\mathbf{x}^*\|_2^2\big) + 16(12um + 3\gamma^2)}{\gamma^2 m}} := \bar{\Lambda}, \quad \text{(B.6)}$$

where in the last inequality we used the fact that the infimum value is attained at $\theta = \min\{\gamma^2/(128u), m/32\}$ and the fact that $1/\theta \leq 128u/\gamma^2 + 32/m$. Therefore, it remains to prove the upper bound of the KL divergence between distributions $\mathbb{P}(\mathbf{z}_K)$ and $\mathbb{P}(\boldsymbol{Z}_{K\eta})$, which can be done by following the standard techniques in [21, 45, 51] to construct a continuous-time Markov process. In particular, based on the update rule in Algorithm 1, we define the following continuous-time interpolation of $(\mathbf{v}_k, \mathbf{x}_k)$

$$\mathrm{d}\widetilde{\boldsymbol{V}}_t = -\gamma\widetilde{\boldsymbol{V}}_t\mathrm{d}t - u\widetilde{\mathbf{G}}_t\mathrm{d}t + \sqrt{2\gamma u}\cdot\mathrm{d}\boldsymbol{B}_t$$
$$\mathrm{d}\widetilde{\boldsymbol{X}}_t = \widetilde{\boldsymbol{V}}_t\mathrm{d}t, \quad \text{(B.7)}$$

where $\widetilde{\mathbf{G}}_t = \sum_{k=0}^{\infty}\mathbf{g}_k\mathbb{1}\{t \in [k\eta, (k+1)\eta)\}$ remains invariant in each interval $[k\eta, (k+1)\eta)$ and $\mathbf{g}_k$ is the semi-stochastic gradient at the $k$-th iteration of Algorithm 1. It can be verified that the distribution of $(\mathbf{v}_k, \mathbf{x}_k)$ is identical to that of $(\widetilde{\boldsymbol{V}}_{k\eta}, \widetilde{\boldsymbol{X}}_{k\eta})$. Integrating (B.7) from 0 to $t$ gives

$$\widetilde{\boldsymbol{V}}_t = \widetilde{\boldsymbol{V}}_0 - \int_0^t \gamma\widetilde{\boldsymbol{V}}_s\mathrm{d}s - \int_0^t u\widetilde{\mathbf{G}}_s\mathrm{d}s + \int_0^t \sqrt{2\gamma u}\cdot\mathrm{d}\boldsymbol{B}_s,$$

$$\widetilde{\boldsymbol{X}}_t = \widetilde{\boldsymbol{X}}_0 + \int_0^t \widetilde{\mathbf{V}}_s\mathrm{d}s.$$

Due to the semi-stochastic gradient $\mathbf{g}_k$, (B.7) does not form a Markov chain since $\widetilde{\mathbf{G}}_s$ contains additional randomness introduced by the stochastic gradient. Nevertheless, Gyöngy [31] showed

that we can use the following Markov chain whose one-time marginal distribution mimics that of $(\widetilde{\boldsymbol{V}}_t, \widetilde{\boldsymbol{X}}_t)$,

$$\widehat{\boldsymbol{V}}_t = \widehat{\boldsymbol{V}}_0 - \int_0^t \gamma \widehat{\boldsymbol{V}}_s \mathrm{d}s - \int_0^t u \widehat{\mathbf{G}}_s \mathrm{d}s + \int_0^t \sqrt{2\gamma u} \cdot \mathrm{d}\boldsymbol{B}_s,$$

$$\widehat{\boldsymbol{X}}_t = \widehat{\boldsymbol{X}}_0 + \int_0^t \widehat{\boldsymbol{V}}_s \mathrm{d}s,$$

where $\widehat{\mathbf{G}}_s = \mathbb{E}[\widetilde{\mathbf{G}}_s | \widetilde{\boldsymbol{V}}_s = \widehat{\boldsymbol{V}}_s]$. Next, we let $\mathbb{P}_t$ denote the probability measure of the point $(\boldsymbol{V}_t, \boldsymbol{X}_t)$ in Hamiltonian Langevin dynamics and $\mathbb{Q}_t$ denote the probability measure of $(\widehat{\boldsymbol{V}}_t, \widehat{\boldsymbol{X}}_t)$. By Girsanov formula [39] we can derive the Radon-Nikodym derivative of $\mathbb{P}_t$ with respect to $\mathbb{Q}_t$ as follows:

$$\frac{\mathrm{d}\mathbb{P}_t}{\mathrm{d}\mathbb{Q}_t} = \exp\left\{ \sqrt{\frac{\gamma u}{2}} \int_0^t \left( \nabla f(\widehat{\boldsymbol{X}}_s) - \widehat{\mathbf{G}}_s \right) \cdot \mathrm{d}\boldsymbol{B}_s - \frac{\gamma u}{4} \int_0^t \left\| \nabla f(\widehat{\boldsymbol{X}}_s) - \widehat{\mathbf{G}}_s \right\|_2^2 \mathrm{d}s \right\}.$$

When we choose $T = K\eta$, it follows that

$$\begin{aligned}
D_{KL}(\mathbb{Q}_T || \mathbb{P}_T) &= \mathbb{E}_{\mathbb{Q}_T}\left[ \log\left( \frac{\mathrm{d}\mathbb{P}_T}{\mathrm{d}\mathbb{Q}_T} \right) \right] \\
&= \frac{\gamma u}{4} \int_0^T \mathbb{E}\left[ \left\| \nabla f(\widehat{\boldsymbol{X}}_s) - \widehat{\mathbf{G}}_s \right\|_2^2 \right] \mathrm{d}s \\
&= \frac{\gamma u}{4} \int_0^T \mathbb{E}\left[ \left\| \nabla f(\widetilde{\boldsymbol{X}}_s) - \widetilde{\mathbf{G}}_s \right\|_2^2 \right] \mathrm{d}s \\
&= \frac{\gamma u}{4} \sum_{k=0}^{K-1} \int_{k\eta}^{(k+1)\eta} \mathbb{E}\left[ \left\| \nabla f(\widetilde{\boldsymbol{X}}_s) - \widetilde{\mathbf{G}}_s \right\|_2^2 \right] \mathrm{d}s, \quad\quad (\mathrm{B}.8)
\end{aligned}$$

where the third equality holds since $\widehat{\boldsymbol{X}}_s$ has the same distribution as $\widetilde{\boldsymbol{X}}_s$. Moreover, note that $\widetilde{\mathbf{G}}_s$ is a step function based on semi-stochastic gradients $\{\mathbf{g}_k\}_{k=1,\ldots,K}$, and equals $\mathbf{g}_k$ when $s \in [k\eta, (k+1)\eta)$ for all $k < K$. Therefore, in the $k$-th interval, i.e., $s \in [k\eta, (k+1)\eta)$, we have

$$\mathbb{E}[\|\nabla f(\widetilde{\boldsymbol{X}}_s) - \widetilde{\mathbf{G}}_s\|_2^2] \le 2\mathbb{E}[\|\nabla f(\widetilde{\boldsymbol{X}}_s) - \nabla f(\mathbf{x}_k)\|_2^2] + 2\mathbb{E}[\|\nabla f(\mathbf{x}_k) - \mathbf{g}_k\|_2^2], \quad\quad (\mathrm{B}.9)$$

where $\mathbf{x}_k = \widetilde{\boldsymbol{X}}_{k\eta}$ is the $k$-th iterate in Algorithm 1. We then upper bound two terms on the R.H.S. of (B.9) separately. Regarding the first term $\mathbb{E}[\|\nabla f(\widetilde{\boldsymbol{X}}_s) - \nabla f(\mathbf{x}_k)\|_2^2]$, Assumption 3.1 implies

$$\mathbb{E}[\|\nabla f(\widetilde{\boldsymbol{X}}_s) - \nabla f(\mathbf{x}_k)\|_2^2] \le M^2 \mathbb{E}[\|\widetilde{\boldsymbol{X}}_s - \widetilde{\boldsymbol{X}}_{k\eta}\|_2^2], \quad\quad (\mathrm{B}.10)$$

where we replaced $\mathbf{x}_k$ with $\widetilde{\boldsymbol{X}}_{k\eta}$. Multiplying $e^{\gamma t}$ to both sides of the first equation in (B.7) yields

$$(\mathrm{d}\widetilde{\boldsymbol{V}}_t + \gamma \widetilde{\boldsymbol{V}}_t \mathrm{d}t) e^{\gamma t} = -u \widetilde{\mathbf{G}}_t e^{\gamma t} \mathrm{d}t + \sqrt{2\gamma u} \cdot e^{\gamma t} \cdot \mathrm{d}\boldsymbol{B}_t.$$

Note that $(\mathrm{d}\widetilde{\boldsymbol{V}}_t + \gamma \widetilde{\boldsymbol{V}}_t dt) e^{\gamma t} = \mathrm{d}(\widetilde{\boldsymbol{V}}_t e^{\gamma t})$, integrating both sides over $t$ from $k\eta$ to $r$ gives

$$\widetilde{\boldsymbol{V}}_r e^{\gamma r} - \widetilde{\boldsymbol{V}}_{k\eta} e^{\gamma k\eta} = \int_{k\eta}^r -u \widetilde{\mathbf{G}}_z e^{\gamma z} \mathrm{d}z + \int_{k\eta}^r \sqrt{2\gamma u} \cdot e^{\gamma z} \cdot \mathrm{d}\boldsymbol{B}_z,$$

which can be further simplified as

$$\widetilde{\boldsymbol{V}}_r = \widetilde{\boldsymbol{V}}_{k\eta} \cdot e^{-\gamma(r-k\eta)} - \int_{k\eta}^r u \widetilde{\mathbf{G}}_z e^{-\gamma(r-z)} \mathrm{d}z + \int_{k\eta}^r \sqrt{2\gamma u} \cdot e^{-\gamma(r-z)} \cdot \mathrm{d}\boldsymbol{B}_z.$$

Thus by the second equation in (B.7) we have

$$\begin{aligned}
\widetilde{\boldsymbol{X}}_s &= \widetilde{\boldsymbol{X}}_{k\eta} + \int_{k\eta}^s \widetilde{\boldsymbol{V}}_r dr \\
&= \widetilde{\boldsymbol{X}}_{k\eta} + \int_{k\eta}^s \left( \widetilde{\boldsymbol{V}}_{k\eta} e^{-\gamma(r-k\eta)} - u\left( \int_{k\eta}^r e^{-\gamma(r-z)} \widetilde{\mathbf{G}}_{k\eta} \mathrm{d}z \right) + \sqrt{2\gamma u} \int_{k\eta}^r e^{-\gamma(r-z)} \mathrm{d}\boldsymbol{B}_z \right) \mathrm{d}r,
\end{aligned}$$

where $\widetilde{\mathbf{G}}_z = \widetilde{\mathbf{G}}_{k\eta}$ for $z \in [k\eta, (k+1)\eta)$ by definition. This further implies that

$$
\begin{aligned}
\|\widetilde{\boldsymbol{X}}_s - \widetilde{\boldsymbol{X}}_{k\eta}\|_2^2 &= \left\| \int_{k\eta}^s \left( \widetilde{\boldsymbol{V}}_{k\eta} e^{-\gamma(r-k\eta)} - u \int_{k\eta}^r e^{-\gamma(r-z)} \widetilde{\mathbf{G}}_{k\eta} \mathrm{d}z + \sqrt{2\gamma u} \int_0^r e^{-\gamma(r-z)} \mathrm{d}\boldsymbol{B}_z \right) \mathrm{d}r \right\|_2^2 \\
&\le 3 \left\| \int_{k\eta}^s \widetilde{\boldsymbol{V}}_{k\eta} e^{-\gamma(r-k\eta)} \mathrm{d}r \right\|_2^2 + 3u^2 \left\| \int_{k\eta}^s \int_{k\eta}^r e^{-\gamma(r-z)} \widetilde{\mathbf{G}}_{k\eta} \mathrm{d}z \mathrm{d}r \right\|_2^2 \\
&\quad + 6\gamma u \left\| \int_{k\eta}^s \int_0^r e^{-\gamma(r-z)} \mathrm{d}\boldsymbol{B}_z \mathrm{d}r \right\|_2^2 \\
&\le 3\eta^2 \|\mathbf{v}_k\|_2^2 + 3u^2\eta^4 \|\mathbf{g}_k\|_2^2 + 6\gamma u \left\| \int_{k\eta}^s \int_0^r e^{-\gamma(r-z)} \mathrm{d}\boldsymbol{B}_z \mathrm{d}r \right\|_2^2,
\end{aligned}
$$

where the second inequality follows from the fact that $(a+b+c)^2 \le 3(a^2+b^2+c^2)$ and the last inequality follows from facts that $s \in [k\eta, (k+1)\eta))$, $\widetilde{\boldsymbol{V}}_{k\eta} = \mathbf{v}_k$, $\widetilde{\mathbf{G}}_{k\eta} = \mathbf{g}_k$ and $e^{-\gamma(r-z)} \le 1$. Moreover, by Lemma B.1, we have

$$
\mathbb{E}\left[ \left\| \int_{k\eta}^s \int_0^r e^{-\gamma(r-z)} \mathrm{d}\boldsymbol{B}_z \mathrm{d}r \right\|_2^2 \right] = \frac{d}{\gamma^2} \left( 2\gamma(s-k\eta) + 4e^{-\gamma(s-k\eta)} - e^{-2\gamma(s-k\eta)} - 3 \right) \le 2d\eta^2,
$$

where we use inequality $1 - x \le e^{-x} \le 1 - x + x^2/2$ for positive $x$ and $0 \le s - k\eta \le \eta$ to get the last inequality. Combining the above analysis and (B.10), we have

$$
\mathbb{E}[\|\nabla f(\widetilde{\boldsymbol{X}}_s) - \nabla f(\mathbf{x}_k)\|_2^2] \le 3M^2\eta^2 \left( \mathbb{E}[\|\mathbf{v}_k\|_2^2] + u^2\eta^2 \mathbb{E}[\|\mathbf{g}_k\|_2^2]/4 + 4\gamma ud \right).
$$

Applying Lemma B.2 and setting $\eta^2 \le \min\{\gamma^2/(4M^2u^2), \gamma^2\bar{\mathcal{E}}/(2G^2u^2)\}$, we have

$$
\mathbb{E}[\|\nabla f(\widetilde{\boldsymbol{X}}_s) - \nabla f(\mathbf{x}_k)\|_2^2] \le 4M^2\gamma^2\eta^2\bar{\mathcal{E}}.
$$

Then in terms of the second term on the R.H.S. of (B.9), we have the following by Lemma B.3,

$$
\mathbb{E}[\|\nabla f(\mathbf{x}_k) - \mathbf{g}_k\|_2^2] \le \frac{4LM^2\gamma^2\eta^2\bar{\mathcal{E}}}{B} + \frac{4M^2\bar{\mathcal{E}}}{B_0} \cdot \mathbb{1}(B_0 < n).
$$

Plugging the above inequalities into (B.9) and further (B.8), we have

$$
\begin{aligned}
D_{KL}(\mathbb{Q}_T \| \mathbb{P}_T) &= \frac{\gamma u}{4} \sum_{k=0}^{K-1} \int_{k\eta}^{(k+1)\eta} \mathbb{E}[\|\nabla f(\widetilde{\boldsymbol{X}}_s) - \widetilde{\mathbf{G}}_s\|_2^2] \\
&\le M^2\gamma^3 u\bar{\mathcal{E}} \left( 1 + \frac{L}{B} \right) K\eta^3 + \frac{M^2\gamma u\bar{\mathcal{E}}K\eta}{B_0} \cdot \mathbb{1}(B_0 < n). \qquad (\text{B.11})
\end{aligned}
$$

Combining (B.5), (B.6) and (B.11) and assuming that $D_{KL}(\mathbb{Q}_T \| \mathbb{P}_T) \le 1$, we get

$$
\mathcal{W}_2(\mathbb{Q}_{K\eta}, \mathbb{P}_{K\eta}) \le 2\bar{\Lambda} \left( M^2\gamma^3 u\bar{\mathcal{E}} \left( 1 + \frac{L}{B} \right) K\eta^3 + \frac{M^2\gamma u\bar{\mathcal{E}}K\eta}{B_0} \cdot \mathbb{1}(B_0 < n) \right)^{1/4},
$$

which completes the proof. $\qquad \square$

## B.2 Proof of Lemma A.2

Now we prove Lemma A.2 which characterizes the exponential mixing rate of the Hamiltonian dynamics (1.1). Our analysis will be built based on the contraction results of Langevin dynamics in [27]. We first lay down some useful lemmas that will be used in our analysis.

The following lemma is a direct implication of Assumption 3.2.

**Lemma B.6.** If $f(\mathbf{x})$ satisfies Assumption 3.2, then for all $\mathbf{x} \in \mathbb{R}^d$ it holds that

$$
\langle \nabla f(\mathbf{x}), \mathbf{x} \rangle / 2 \ge \lambda \left( f(\mathbf{x}) + u^{-1}\gamma^2 \|\mathbf{x}\|_2^2/4 \right) - A, \qquad (\text{B.12})
$$

where $\lambda$ and $A$ are parameters defined as

$$
\lambda = \frac{2m}{4M + u^{-1}\gamma^2} \quad \text{and} \quad A = \frac{2m(f(\mathbf{x}^*) + M\|\mathbf{x}^*\|_2^2)}{4M + u^{-1}\gamma^2} + \frac{b}{2}. \qquad (\text{B.13})
$$

Before we present the contraction results provided in [27], we first define a semi-metric $\mathcal{W}_\rho(\cdot,\cdot)$. For any concatenated vectors $(\mathbf{x},\mathbf{v}),(\mathbf{x}',\mathbf{v}') \in \mathbb{R}^{2d}$ (equivalently, $\mathbf{x},\mathbf{v},\mathbf{x}',\mathbf{v}' \in \mathbb{R}^d$), we define

$$
\begin{aligned}
r((\mathbf{x},\mathbf{v}),(\mathbf{x}',\mathbf{v}')) &= \alpha \|\mathbf{x}-\mathbf{x}'\|_2 + \|\mathbf{x}-\mathbf{x}'+\gamma^{-1}(\mathbf{v}-\mathbf{v}')\|_2, \\
\rho((\mathbf{x},\mathbf{v}),(\mathbf{x}',\mathbf{v}')) &= h(r((\mathbf{x},\mathbf{v}),(\mathbf{x}',\mathbf{v}')))(1+\theta\mathcal{V}(\mathbf{x},\mathbf{v})+\theta\mathcal{V}(\mathbf{x}',\mathbf{v}')),
\end{aligned}
\tag{B.14}
$$

where $\alpha,\theta \in (0,\infty)$ are constants. $h : [0,\infty) \to [0,\infty)$ (1) is a continuous, non-decreasing concave function which is $C^2$ continuous on $(0,R_1)$ for some constant $R_1 > 0$; (2) is a constant function on $[0,\infty)$; (3) and satisfies $h(0) = 0$, $h'_+(0) = 1$ and $h'_-(R_1) > 0$. $\mathcal{V} : \mathbb{R}^{2d} \to \mathbb{R}$ is defined as follows

$$
\mathcal{V}(\mathbf{x},\mathbf{v}) = f(\mathbf{x}) + \frac{\gamma^2}{4u}\big(\|\mathbf{x}+\gamma^{-1}\mathbf{v}\|_2^2 + \|\gamma^{-1}\mathbf{v}\|_2^2 - \lambda\|\mathbf{x}\|_2^2\big),
$$

where $\gamma, u$ are the parameter of dynamics in (1.1), $\lambda$ is defined in (B.13). For any two probability measures $\mu$ and $\nu$, we define

$$
\mathcal{W}_\rho(\mu,\nu) = \inf_{\zeta \in \Gamma(\mu,\nu)} \int \rho((\mathbf{x},\mathbf{v}),(\mathbf{x}',\mathbf{v}'))\mathrm{d}\zeta((\mathbf{x},\mathbf{v}),(\mathbf{x}',\mathbf{v}')),
\tag{B.15}
$$

where the infimum is over all couplings of $\mu$ and $\nu$. As is pointed out by Eberle et al. [27], $\mathcal{W}_\rho$ may not necessarily be a metric and thus triangle inequality does not hold. Therefore, we call $\mathcal{W}_\rho$ a semi-metric.

Recall the solution of Hamiltonian dynamics in Lemma B.1. We use $\mathcal{L}_t$ to denote the operator of integration on the dynamics from time 0 to $t$. That is, $\mathcal{L}_t\mathbf{V}_0 = \mathbf{V}_t$ and $\mathcal{L}_t\mathbf{X}_0 = \mathbf{X}_t$ denote the velocity and the position of the random process. Suppose the initial point $\mathbf{Z}_0 = (\mathbf{X}_0^\top,\mathbf{V}_0^\top)^\top \in \mathbb{R}^{2d}$ follows a distribution $\mu$. Then with a slight abuse of notation, we also use $\mathcal{L}_t\mu$ to denote the distribution of $\mathbf{Z}_t = (\mathbf{X}_t^\top,\mathbf{V}_t^\top)^\top$. Built on the above preliminaries and notations, the following lemma is about the contraction of Hamiltonian dynamics in terms of semi-metric $\mathcal{W}_\rho$.

**Lemma B.7** (Theorem 2.3 and Corollary 2.6 in [27]). Suppose Assumptions 3.1 and 3.2 hold and thus (B.12) is true. There exist constants $\alpha,\theta > 0$ and a continuous non-decreasing concave function $h : [0,\infty) \to [0,\infty)$ as required in (B.14) such that for all probability measures $\mu,\nu$, it holds that

$$
\mathcal{W}_2(\mathcal{L}_t\mu,\mathcal{L}_t\nu) \leq C_0\sqrt{\mathcal{W}_\rho(\mu,\nu)}e^{-\mu_* t}
$$

for all $t \geq 0$, where $\mu_*$ is a lower bound of the spectral gap of Markov chain (1.1) and satisfies

$$
\begin{aligned}
\mu_* &= \frac{1}{768\gamma e^\Lambda}\min\{\lambda M u e^\Lambda, \Lambda^{1/2}Mu, \gamma\Lambda^{1/2}\}, \\
\Lambda &= \frac{12(1+2\alpha+2\alpha^2)(d+A)Mu}{5\gamma^2\lambda(1-2\lambda)}, \\
C_0 &= \frac{\sqrt{2}e^{1+\Lambda/2}}{\min\{1,\alpha\}}\max\left\{1, \frac{2\sqrt{2+4\alpha+4\alpha^2}(d+A)^{1/2}u^{1/2}\gamma^{-1/2}\mu_*^{-1/2}}{\min\{1,(8\Lambda/M)^{1/4}\}}\right\},
\end{aligned}
\tag{B.16}
$$

$\gamma, u$ are the parameters in dynamics (1.1) and $\lambda, A$ are defined in (B.13).

In particular, for Lemma B.7, the function $h$ in the definition of semi-metric $\mathcal{W}_\rho$ in (B.15) is chosen as follows:

$$
h(r) = \int_0^{r\wedge R_1} \phi(s)g(s)\mathrm{d}s,
$$

where $R_1 = \sqrt{8\Lambda/M}$ and the auxiliary functions are defined as

$$
\phi(s) = e^{-\frac{(1+\eta)Ms^2}{8} - \frac{\gamma^2 s^2 \max\{1,1/(2\alpha)\}}{2u}}, \quad \Phi(s) = \int_0^s \phi(x)\mathrm{d}x,
$$

$$
g(s) = 1 - \frac{9\lambda^*\gamma}{4u}\int_0^s \Phi(x)\phi(x)^{-1}\mathrm{d}x.
$$

Now we are ready to complete the proof of Lemma A.2.

*Proof of Lemma A.2.* By (2.11) in [27], we know that the function $h(r)$ is upper bounded by $R_1 = \sqrt{8\Lambda/M}$ defined in Lemma B.7. Thus, the distance function $\rho((\mathbf{x}, \mathbf{v}), (\mathbf{x}', \mathbf{v}'))$ can be bounded as

$$\rho((\mathbf{x}, \mathbf{v}), (\mathbf{x}', \mathbf{v}')) \leq R_1(1 + \theta\mathcal{V}(\mathbf{x}, \mathbf{v}) + \theta\mathcal{V}(\mathbf{x}', \mathbf{v}')).$$

Let $\mu_0$ denote the distribution of $(\mathbf{x}_0, \mathbf{v}_0)$. It follows that

$$
\begin{aligned}
\mathcal{W}_\rho(\mu_0, \pi_{\mathbf{z}}) &= \inf_{\zeta \in \Gamma(\mu_0, \pi_{\mathbf{z}})} \int \rho((\mathbf{x}_0, \mathbf{v}_0), (\mathbf{x}_\pi, \mathbf{v}_\pi)) d\zeta((\mathbf{x}_0, \mathbf{v}_0), (\mathbf{x}^\pi, \mathbf{v}^\pi)) \\
&\leq R_1\big(1 + \theta\mathbb{E}[\mathcal{V}(\mathbf{x}_0, \mathbf{v}_0)] + \theta\mathbb{E}[\mathcal{V}(\mathbf{x}^\pi, \mathbf{v}^\pi)]\big).
\end{aligned}
\tag{B.17}
$$

Moreover, recall the definition of function $\mathcal{V}(\mathbf{x}, \mathbf{v})$ we have

$$
\begin{aligned}
\mathcal{V}(\mathbf{x}, \mathbf{v}) &= f(\mathbf{x}) + \frac{\gamma^2}{4u}\big(\|\mathbf{x} + \gamma^{-1}\mathbf{v}\|_2^2 + \|\gamma^{-1}\mathbf{v}\|_2^2 - \lambda\|\mathbf{x}\|_2^2\big) \\
&\leq f(\mathbf{x}^*) + \frac{M}{2}\|\mathbf{x} - \mathbf{x}^*\|_2^2 + \frac{\gamma^2}{4u}\big(\|\mathbf{x} + \gamma^{-1}\mathbf{v}\|_2^2 + \|\gamma^{-1}\mathbf{v}\|_2^2 - \lambda\|\mathbf{x}\|_2^2\big) \\
&\leq f(\mathbf{x}^*) + \left(M + \frac{\gamma^2(2 - \lambda)}{4u}\right)\|\mathbf{x}\|_2^2 + M\|\mathbf{x}^*\|_2^2 + \frac{3\|\mathbf{v}\|_2^2}{4u},
\end{aligned}
$$

where the first inequality comes from the smoothness of $f$ (Assumption 3.1) and the second inequality is due to the fact that $(a + b)^2 \leq 2a^2 + 2b^2$ for any $a, b$. Note that the stationary distribution $\pi$ is proportional to the function $e^{-f(\mathbf{x}) - \|\mathbf{v}\|_2^2/(2u)}$. As is shown in [45] (Section 3.5, equation (3.19)), we know that

$$\mathbb{E}[\|\mathbf{x}^\pi\|_2^2] \leq \frac{b + d}{m}.$$

In addition, since the marginal distribution of $\mathbf{v}^\pi$ is a $d$ dimensional Gaussian distribution, we have $\mathbb{E}[\|\mathbf{v}^\pi\|_2^2] = du$. Plugging these into (B.17), we have

$$
\begin{aligned}
\mathcal{W}_\rho(\mu_0, \pi_{\mathbf{z}}) &\leq R_1\big(1 + \theta\mathbb{E}[\mathcal{V}(\mathbf{x}_0, \mathbf{v}_0)] + \theta\mathbb{E}[\mathcal{V}(\mathbf{x}^\pi, \mathbf{v}^\pi)]\big) \\
&\leq R_1\left(1 + \theta\left(\mathcal{V}(\mathbf{x}_0, \mathbf{v}_0) + f(\mathbf{x}^*) + \frac{(4Mu + \gamma^2(2 - \lambda))(b + d)}{4um} + d + M\|\mathbf{x}^*\|_2^2\right)\right).
\end{aligned}
$$

Moreover, note the fact that $\mathbf{x}_0 = \mathbf{v}_0 = \mathbf{0}$. We further obtain

$$\mathcal{W}_\rho(\mu_0, \pi_{\mathbf{z}}) \leq R_1\left(1 + \theta\left(2f(\mathbf{x}^*) + \frac{(4Mu + \gamma^2(2 - \lambda))(b + d)}{4um} + d + 2M\|\mathbf{x}^*\|_2^2\right)\right) := \Theta.$$

For the stationary distribution $\pi_{\mathbf{z}}$, it is invariant under the Hamiltonian dynamics, i.e., $\mathcal{L}_t \pi_{\mathbf{z}} = \pi_{\mathbf{z}}$ for any $t \geq 0$. Note that $\mathcal{L}_t \mu_0 = \mathbb{P}(\mathbf{Z}_t)$ by definition. By Lemma B.7, we have

$$
\begin{aligned}
\mathcal{W}_2\big(\mathbb{P}(\mathbf{Z}_t), \pi_{\mathbf{z}}\big) &\leq C_0\sqrt{\mathcal{W}_\rho\big(\mathbb{P}(\mathbf{Z}_0), \pi_{\mathbf{z}}\big)}e^{-\mu_* t} \\
&\leq C_0\Theta^{1/2}e^{-\mu_* t},
\end{aligned}
$$

where $\mu_*$ and $C_0$ are defined in (B.16) in Lemma B.7. Let $\Gamma_0 = C_0\Theta^{1/2}$, it can be seen that both $\Gamma_0$ and $1/\mu_*$ are in the order of $\exp(\widetilde{O}(d))$. This completes the proof. $\square$

# C   Proof of technical lemmas in Appendix B

In this section, we prove the technical lemmas used in the proof of our key lemmas.

## C.1   Proof of Lemma B.2

We first present the following lemma on upper bound of the gradient norm, which is a straightforward implication of the smoothness of $f$.

**Lemma C.1.** Under Assumption 3.1, for all $\mathbf{x} \in \mathbb{R}^d$ and $i \in [n]$, it holds that

$$\|\nabla f_i(\mathbf{x})\|_2 \leq M\|\mathbf{x}\|_2 + G,$$

where $G = \max_{i \in [n]} \|\nabla f_i(\mathbf{0})\|_2$ is a constant.

**Lemma C.2.** Under Assumption 3.1, let $k = jm + l$, it holds that

$$\mathbb{E}[\|\mathbf{g}_k - \nabla f(\mathbf{x}_k)\|_2^2] \leq \frac{M^2}{B} \sum_{s=jm}^{jm+l} \mathbb{E}[\|\mathbf{x}_{s+1} - \mathbf{x}_s\|_2^2] + \frac{2}{B_0} \mathbb{E}[\|\mathbf{x}_{jm}\|_2^2 + G^2] \cdot \mathbb{1}(B_0 < n),$$

where $G$ follows the same definition in Lemma C.1.

*Proof of Lemma B.2.* Recall the Lyapunov function defined in (B.3), we have

$$\mathcal{E}(\mathbf{x}_{k+1}, \mathbf{v}_{k+1}) = \|\mathbf{x}_{k+1}\|_2^2 + \|\mathbf{x}_{k+1} + 2\mathbf{v}_{k+1}/\gamma\|_2^2 + 8u(f(\mathbf{x}_{k+1}) - f(\mathbf{x}^*))/\gamma^2. \tag{C.1}$$

By Assumption 3.1, it holds that

$$f(\mathbf{x}_{k+1}) - f(\mathbf{x}^*) \leq f(\mathbf{x}_k) + \langle \nabla f(\mathbf{x}_k), \mathbf{x}_{k+1} - \mathbf{x}_k \rangle + M\|\mathbf{x}_{k+1} - \mathbf{x}_k\|_2^2/2 - f(\mathbf{x}^*). \tag{C.2}$$

For the first two terms in (C.1), we have

$$\|\mathbf{x}_{k+1}\|_2^2 = \|\mathbf{x}_k\|_2^2 + 2\langle \mathbf{x}_k, \mathbf{x}_{k+1} - \mathbf{x}_k \rangle + \|\mathbf{x}_{k+1} - \mathbf{x}_k\|_2^2,$$

$$\|\mathbf{x}_{k+1} + 2\mathbf{v}_{k+1}/\gamma\|_2^2 = \|\mathbf{x}_k + 2\mathbf{v}_k/\gamma\|_2^2 + 2\langle \mathbf{x}_k + 2\mathbf{v}_k/\gamma, \mathbf{x}_{k+1} - \mathbf{x}_k + 2(\mathbf{v}_{k+1} - \mathbf{v}_k)/\gamma \rangle$$
$$+ \|\mathbf{x}_{k+1} - \mathbf{x}_k + 2(\mathbf{v}_{k+1} - \mathbf{v}_k)/\gamma\|_2^2,$$

Substituting the above two equations and (C.2) into (C.1) yields

$$\mathbb{E}[\mathcal{E}(\mathbf{x}_{k+1}, \mathbf{v}_{k+1})]$$

$$\leq \mathbb{E}[\mathcal{E}(\mathbf{x}_k, \mathbf{v}_k)] + 4\mathbb{E}[\langle \mathbf{x}_k, \mathbf{x}_{k+1} - \mathbf{x}_k \rangle] + \frac{4}{\gamma}\mathbb{E}[\langle \mathbf{x}_k, \mathbf{v}_{k+1} - \mathbf{v}_k \rangle] + \frac{4}{\gamma}\mathbb{E}[\langle \mathbf{v}_k, \mathbf{x}_{k+1} - \mathbf{x}_k \rangle]$$

$$+ \frac{8}{\gamma^2}\mathbb{E}[\langle \mathbf{v}_k, \mathbf{v}_{k+1} - \mathbf{v}_k \rangle] + \frac{8u}{\gamma^2}\mathbb{E}[\langle \nabla f(\mathbf{x}_k), \mathbf{x}_{k+1} - \mathbf{x}_k \rangle + M/2\|\mathbf{x}_{k+1} - \mathbf{x}_k\|_2^2]$$

$$+ \mathbb{E}[\|\mathbf{x}_{k+1} - \mathbf{x}_k\|_2^2] + \mathbb{E}[\|\mathbf{x}_{k+1} - \mathbf{x}_k + 2(\mathbf{v}_{k+1} - \mathbf{v}_k)/\gamma\|_2^2]. \tag{C.3}$$

Next, we need to upper bound inner products terms $\langle \mathbf{x}_k, \mathbf{x}_{k+1} - \mathbf{x}_k \rangle$, $\langle \mathbf{x}_k, \mathbf{v}_{k+1} - \mathbf{v}_k \rangle$, $\langle \mathbf{v}_k, \mathbf{x}_{k+1} - \mathbf{x}_k \rangle$, and $\langle \mathbf{v}_k, \mathbf{v}_{k+1} - \mathbf{v}_k \rangle$ respectively. Recall the update formula of Algorithm 1 as follows,

$$\mathbf{v}_{k+1} = \mathbf{v}_k e^{-\gamma\eta} - \frac{u(1 - e^{-\gamma\eta})}{\gamma}\mathbf{g}_k + \boldsymbol{\epsilon}_k^v,$$

$$\mathbf{x}_{k+1} = \mathbf{x}_k + \frac{1 - e^{-\gamma\eta}}{\gamma}\mathbf{v}_k + \frac{u(\gamma\eta + e^{-\gamma\eta} - 1)}{\gamma^2}\mathbf{g}_k + \boldsymbol{\epsilon}_k^x. \tag{C.4}$$

Note that $\boldsymbol{\epsilon}_k^v$ and $\boldsymbol{\epsilon}_k^x$ are zero mean and independent of $\mathbf{v}_k, \mathbf{x}_k$ and $\mathbf{g}_k$. Then we have

$$\mathbb{E}[\langle \mathbf{x}_k, \mathbf{x}_{k+1} - \mathbf{x}_k \rangle] = \frac{1 - e^{-\gamma\eta}}{\gamma}\mathbb{E}[\langle \mathbf{x}_k, \mathbf{v}_k \rangle] + \frac{u(\gamma\eta + e^{-\gamma\eta} - 1)}{\gamma^2}\mathbb{E}[\langle \mathbf{x}_k, \mathbf{g}_k \rangle],$$

$$\mathbb{E}[\langle \mathbf{x}_k, \mathbf{v}_{k+1} - \mathbf{v}_k \rangle] = -(1 - e^{-\gamma\eta})\mathbb{E}[\langle \mathbf{x}_k, \mathbf{v}_k \rangle] - \frac{u(1 - e^{-\gamma\eta})}{\gamma}\mathbb{E}[\langle \mathbf{x}_k, \mathbf{g}_k \rangle],$$

$$\mathbb{E}[\langle \mathbf{v}_k, \mathbf{x}_{k+1} - \mathbf{x}_k \rangle] = \frac{1 - e^{-\gamma\eta}}{\gamma}\mathbb{E}[\|\mathbf{v}_k\|_2^2] + \frac{u(\gamma\eta + e^{-\gamma\eta} - 1)}{\gamma^2}\mathbb{E}[\langle \mathbf{v}_k, \mathbf{g}_k \rangle],$$

$$\mathbb{E}[\langle \mathbf{v}_k, \mathbf{v}_{k+1} - \mathbf{v}_k \rangle] = -(1 - e^{-\gamma\eta})\mathbb{E}[\|\mathbf{v}_k\|_2^2] - \frac{u(1 - e^{-\gamma\eta})}{\gamma}\mathbb{E}[\langle \mathbf{v}_k, \mathbf{g}_k \rangle].$$

Plugging the above bounds for inner products and (C.4) into (C.3) yields

$$\mathbb{E}[\mathcal{E}(\mathbf{x}_{k+1}, \mathbf{v}_{k+1})]$$

$$\leq \mathbb{E}[\mathcal{E}(\mathbf{x}_k, \mathbf{v}_k)] - \frac{4u(2 - \gamma\eta - 2e^{-\gamma\eta})}{\gamma^2}\mathbb{E}[\langle \mathbf{x}_k, \mathbf{g}_k \rangle] - \frac{4(1 - e^{-\gamma\eta})}{\gamma^2}\mathbb{E}[\|\mathbf{v}_k\|_2^2]$$

$$+ \frac{4u(\gamma\eta + e^{-\gamma\eta} - 1)}{\gamma^3}\mathbb{E}[\langle \mathbf{v}_k, \mathbf{g}_k \rangle] + \frac{8u(1 - e^{-\gamma\eta})}{\gamma^3}\mathbb{E}[\langle \mathbf{v}_k, \nabla f(\mathbf{x}_k) - \mathbf{g}_k \rangle]$$

$$+ \frac{8u^2(\gamma\eta + e^{-\gamma\eta} - 1)}{\gamma^4}\mathbb{E}[\langle \nabla f(\mathbf{x}_k), \mathbf{g}_k \rangle] + \left(\frac{4Mu}{\gamma^2} + 3\right)\mathbb{E}[\|\mathbf{x}_{k+1} - \mathbf{x}_k\|_2^2]$$

$$+ \frac{8}{\gamma^2}\mathbb{E}[\|\mathbf{v}_{k+1} - \mathbf{v}_k\|_2^2]. \tag{C.5}$$

By Assumption 3.2, we know that $\langle \mathbf{x}_k, \nabla f(\mathbf{x}_k)\rangle \geq m\|\mathbf{x}_k\|_2^2 - b$. We then assume $\eta \leq 1/(8\gamma)$ and use the inequality $-x \leq e^{-x} - 1 \leq x^2/2 - x$ for any $x \geq 0$, it follows that

$$\frac{4u(2 - \gamma\eta - 2e^{-\gamma\eta})}{\gamma^2}\mathbb{E}[\langle \mathbf{x}_k, \mathbf{g}_k\rangle]$$

$$= \frac{4u(2 - \gamma\eta - 2e^{-\gamma\eta})}{\gamma^2}\left[\mathbb{E}[\langle \mathbf{x}_k, \nabla f(\mathbf{x}_k)\rangle] + \mathbb{E}[\langle \mathbf{x}_k, \mathbf{g}_k - \nabla f(\mathbf{x})\rangle]\right]$$

$$\geq \frac{4u(2 - \gamma\eta - 2e^{-\gamma\eta})}{\gamma^2}\left(m\|\mathbf{x}_k\|_2^2 - b\right) - \frac{4u(2 - \gamma\eta - 2e^{-\gamma\eta})}{\gamma^2}\left(\frac{1}{8}\mathbb{E}[\|\mathbf{x}_k\|_2^2] + 2\mathbb{E}[\|\mathbf{g}_k - \nabla f(\mathbf{x}_k)\|_2^2]\right)$$

$$\geq \frac{3mu\eta}{\gamma}\|\mathbf{x}_k\|_2^2 - \frac{4u\eta b}{\gamma} - \frac{8u\eta}{\gamma}\mathbb{E}[\|\mathbf{g}_k - \nabla f(\mathbf{x}_k)\|_2^2],$$

where the first inequality is by Young's inequality and the last one is based on the inequality $\gamma\eta - (\gamma\eta)^2 \leq 2 - \gamma\eta - 2e^{-\gamma\eta} \leq \gamma\eta$. Similarly, by Young's inequality, we also have

$$\frac{8u(1 - e^{-\gamma})}{\gamma^3}\mathbb{E}[\langle \mathbf{v}_k, \nabla f(\mathbf{x}_k) - \mathbf{g}_k\rangle] \leq \frac{8u(1 - e^{-\gamma\eta})}{\gamma^3}\left[\frac{\gamma}{8u}\mathbb{E}[\|\mathbf{v}_k\|_2^2] + \frac{2u}{\gamma}\mathbb{E}[\|\nabla f(\mathbf{x}_k) - \mathbf{g}_k\|_2^2]\right]$$

$$\leq \frac{1 - e^{-\gamma\eta}}{\gamma^2}\mathbb{E}[\|\mathbf{v}_k\|_2^2] + \frac{16u^2\eta}{\gamma^3}\mathbb{E}[\|\nabla f(\mathbf{x}_k) - \mathbf{g}_k\|_2^2].$$

Then again by Young's inequalities $\mathbb{E}[\langle \mathbf{v}_k, \mathbf{g}_k\rangle] \leq 1/2\mathbb{E}[\|\mathbf{v}_k\|_2^2] + 1/2\mathbb{E}[\|\mathbf{g}_k\|_2^2]$ and $\mathbb{E}[\langle \nabla f(\mathbf{x}_k), \mathbf{g}_k\rangle] \leq 1/2\mathbb{E}[\|\nabla f(\mathbf{x}_k)\|_2^2] + 1/2\mathbb{E}[\|\mathbf{g}_k\|_2^2]$, (C.5) can be further simplified as

$$\mathbb{E}[\mathcal{E}(\mathbf{x}_{k+1}, \mathbf{v}_{k+1})]$$

$$\leq \mathbb{E}[\mathcal{E}(\mathbf{x}_k, \mathbf{v}_k)] - \frac{3um\eta}{\gamma}\mathbb{E}[\|\mathbf{x}_k\|_2^2] + \frac{4u\eta b}{\gamma} - \frac{3(1 - e^{-\gamma\eta}) - u\gamma\eta^2}{\gamma^2}\mathbb{E}[\|\mathbf{v}_k\|_2^2]$$

$$+ \frac{8u\gamma^2\eta + 16u^2\eta}{\gamma^3}\mathbb{E}[\|\nabla f(\mathbf{x}_k) - \mathbf{g}_k\|_2^2] + \frac{(2u + \gamma)u\eta^2}{\gamma^2}\mathbb{E}[\|\mathbf{g}_k\|_2^2] + \frac{2u^2\eta^2}{\gamma^2}\mathbb{E}[\|\nabla f(\mathbf{x}_k)\|_2^2]$$

$$+ \left(\frac{4Mu}{\gamma^2} + 3\right)\mathbb{E}[\|\mathbf{x}_{k+1} - \mathbf{x}_k\|_2^2] + \frac{8}{\gamma^2}\mathbb{E}[\|\mathbf{v}_{k+1} - \mathbf{v}_k\|_2^2], \tag{C.6}$$

where we use the inequality $-x \leq e^{-x} - 1 \leq x^2/2 - x$ again. We then focus on bounding terms $\mathbb{E}[\|\mathbf{x}_{k+1} - \mathbf{x}_k\|_2^2]$ and $\mathbb{E}[\|\mathbf{v}_{k+1} - \mathbf{v}_k\|_2^2]$. According to (C.4), we have

$$\mathbb{E}[\|\mathbf{x}_{k+1} - \mathbf{x}_k\|_2^2] = \mathbb{E}\left[\left\|\frac{1 - e^{-\gamma\eta}}{\gamma}\mathbf{v}_k + \frac{u(\gamma\eta + e^{-\gamma\eta} - 1)}{\gamma^2}\mathbf{g}_k\right\|_2^2\right] + \mathbb{E}[\|\boldsymbol{\epsilon}_k^x\|_2^2]$$

$$\leq 2\eta^2\mathbb{E}[\|\mathbf{v}_k\|_2^2] + \frac{u^2\eta^4}{2}\mathbb{E}[\|\mathbf{g}_k\|_2^2] + \mathbb{E}[\|\boldsymbol{\epsilon}_k^x\|_2^2], \tag{C.7}$$

where the first equation is due to the independence between $\boldsymbol{\epsilon}_k^x$ and $\mathbf{v}_k, \mathbf{g}_k$, and the inequality come from the fact that $-x \leq e^{-x} - 1 \leq x^2/2 - x$ and Young's inequality to (C.4). Similarly, we also have

$$\mathbb{E}[\|\mathbf{v}_{k+1} - \mathbf{v}_k\|_2^2] \leq 2\gamma^2\eta^2\mathbb{E}[\|\mathbf{v}_k\|_2^2] + 2u^2\eta^2\mathbb{E}[\|\mathbf{g}_k\|_2^2] + \mathbb{E}[\|\boldsymbol{\epsilon}_k^v\|_2^2]. \tag{C.8}$$

Furthermore, by (2.3) it can be easily verified that $\mathbb{E}[\|\boldsymbol{\epsilon}_k^v\|_2^2] \leq 2\gamma ud\eta$ and $\mathbb{E}[\|\boldsymbol{\epsilon}_k^x\|_2^2] \leq 2ud\eta^2$. Plugging (C.7) and (C.8) into (C.6) gives

$$\mathbb{E}[\mathcal{E}(\mathbf{x}_{k+1}, \mathbf{v}_{k+1})]$$

$$\leq \mathbb{E}[\mathcal{E}(\mathbf{x}_k, \mathbf{v}_k)] - \frac{3um\eta^2}{\gamma}\mathbb{E}[\|\mathbf{x}_k\|_2^2] - \frac{3(1 - e^{-\gamma\eta}) - \eta^2(8Mu + u\gamma + 22\gamma^2)}{\gamma^2}\mathbb{E}[\|\mathbf{v}_k\|_2^2]$$

$$+ \frac{36u^2\eta^2 + 2\gamma u\eta^2 + (4Mu + 3\gamma^2)\eta^4}{2\gamma^2}\mathbb{E}[\|\mathbf{g}_k\|_2^2] + \frac{2u^2\eta^2}{\gamma^2}\mathbb{E}[\|\nabla f(\mathbf{x}_k)\|_2^2]$$

$$+ \frac{8u\eta(\gamma^2 + 2u)}{\gamma^3}\mathbb{E}[\|\nabla f(\mathbf{x}_k) - \mathbf{g}_k\|_2^2] + \frac{(8Mu + 6\gamma^2)ud\eta^2 + 4(4d + b)u\gamma\eta}{\gamma^2}, \tag{C.9}$$

where we the fact that $-x \leq e^{-x} - 1 \leq x^2/2 - x$. Note that $1 - \exp(x) \geq 3x/4$ when $x \leq 1/2$. Thus, we set

$$\eta \leq \min \left\{ \frac{\gamma}{4(8Mu + u\gamma + 22\gamma^2)}, \sqrt{\frac{4u^2}{4Mu + 3\gamma^2}}, \frac{6\gamma bu}{(4Mu + 3\gamma^2)d} \right\},$$

and obtain the following according to (C.9),

$$\mathbb{E}[\mathcal{E}(\mathbf{x}_{k+1}, \mathbf{v}_{k+1})] \leq \mathbb{E}[\mathcal{E}(\mathbf{x}_k, \mathbf{v}_k)] - \frac{3um\eta}{\gamma}\mathbb{E}[\|\mathbf{x}_k\|_2^2] - \frac{2\eta}{\gamma}\mathbb{E}[\|\mathbf{v}_k\|_2^2] + \frac{(20u + \gamma)u\eta^2}{\gamma^2}\mathbb{E}[\|\mathbf{g}_k\|_2^2]$$
$$+ \frac{2u^2\eta^2}{\gamma^2}\mathbb{E}[\|\nabla f(\mathbf{x}_k)\|_2^2] + \frac{8u\eta(\gamma^2 + 2u)}{\gamma^3}\mathbb{E}[\|\nabla f(\mathbf{x}_k) - \mathbf{g}_k\|_2^2] + \frac{16(d + b)u\eta}{\gamma}.$$

$$\text{(C.10)}$$

We complete the proof via induction. In particular, we aim to prove the following

$$\mathbb{E}[\mathcal{E}(\mathbf{x}_k, \mathbf{v}_k)] \leq \bar{\mathcal{E}} = \mathcal{E}(\mathbf{x}_0, \mathbf{v}_0) + \frac{8Mu[20(d + b) + m\|\mathbf{x}^*\|_2^2]}{\gamma^2 m} + \frac{G^2}{M^2}, \qquad \text{(C.11)}$$

$$\mathbb{E}[\|\mathbf{g}_k\|_2^2] \leq 2M^2\bar{\mathcal{E}} + 2\mathbb{E}[\|\nabla f(\mathbf{x}_k)\|_2^2].$$

First, it is easy to verify that (C.11) holds for $(\mathbf{x}_0, \mathbf{v}_0)$. Then we assume it holds for all $(\mathbf{x}_s, \mathbf{v}_s)$ with $s \leq k$, and prove it remains true for $(\mathbf{x}_{k+1}, \mathbf{v}_{k+1})$. It is worthy noting that by (B.4), we have $\mathbb{E}[\|\mathbf{x}_s\|_2^2] \leq \bar{\mathcal{E}}$ and $\mathbb{E}[\|\mathbf{v}_s\|_2^2] \leq \gamma^2\bar{\mathcal{E}}/2$ for all $s \leq k$.

**Induction for $\mathbb{E}[\|\mathbf{g}_{k+1}\|_2^2]$:** Regarding $\mathbb{E}[\|\mathbf{g}_{k+1}\|_2^2]$, we aim to show that $\mathbb{E}[\|\mathbf{g}_{k+1}\|_2^2] \leq 2M^2\bar{\mathcal{E}} + 2\mathbb{E}[\|\nabla f(\mathbf{x}_{k+1})\|_2^2]$. By Lemma C.2, we have

$$\mathbb{E}[\|\mathbf{g}_{k+1} - \nabla f(\mathbf{x}_{k+1})\|_2^2] \leq \frac{M^2}{B}\sum_{s=jm}^{jm+l}\mathbb{E}[\|\mathbf{x}_{s+1} - \mathbf{x}_s\|_2^2] + \frac{2}{B_0}\left(\mathbb{E}[M^2\|\mathbf{x}_{jm}\|_2^2] + G^2\right) \cdot \mathbb{1}(B_0 < n)$$

$$\leq \frac{M^2}{B_0}\sum_{s=jm}^{k}\left[2\eta^2\mathbb{E}[\|\mathbf{v}_s\|_2^2] + \frac{u^2\eta^4}{2}\mathbb{E}[\|\mathbf{g}_s\|_2^2] + \mathbb{E}[\|\boldsymbol{\epsilon}_s^x\|_2^2]\right]$$

$$+ \frac{2}{B}\left(\mathbb{E}[M^2\|\mathbf{x}_{jm}\|_2^2] + G^2\right). \qquad \text{(C.12)}$$

Note that by the induction assumption, Lemma C.1 and Young's inequality, we have

$$\mathbb{E}[\|\nabla f(\mathbf{x}_s)\|_2^2] \leq 2M^2\mathbb{E}[\|\mathbf{x}_s\|_2^2] + 2G^2 \leq 2M^2\bar{\mathcal{E}} + 2G^2,$$

for all $s \leq k$, which implies $\mathbb{E}[\|\mathbf{g}_s\|_2^2] \leq 6M^2\bar{\mathcal{E}} + 4G^2$ for all $s \leq k$. In addition we have $\mathbb{E}[\|\mathbf{x}_{jm}\|_2^2] \leq \bar{\mathcal{E}}$ since $jm \leq k$. Therefore, we have

$$\mathbb{E}[\|\mathbf{g}_{k+1} - \nabla f(\mathbf{x}_{k+1})\|_2^2] \leq \frac{M^2}{B}\sum_{s=jm}^{k}\left[2\eta^2\mathbb{E}[\|\mathbf{v}_s\|_2^2] + \frac{u^2\eta^4}{2}\mathbb{E}[\|\mathbf{g}_s\|_2^2] + \mathbb{E}[\|\boldsymbol{\epsilon}_s^x\|_2^2]\right] + \frac{2}{B_0}(M^2\bar{\mathcal{E}} + G^2)$$

$$\leq \frac{LM^2}{B}\left[\gamma^2\eta^2\bar{\mathcal{E}} + u^2(3M\bar{\mathcal{E}} + 2G^2)\eta^4 + 2ud\eta^2\right] + \frac{4M^2\bar{\mathcal{E}}}{B_0} \cdot \mathbb{1}(B_0 < n),$$

where we use the fact that $G^2 \leq M^2\bar{\mathcal{E}}$. Let $\eta^2 \leq \min\{\gamma^2/(6M^2u^2), \gamma^2\bar{\mathcal{E}}/(4G^2u^2)\}$, and use the fact that $u \leq \gamma^2\bar{\mathcal{E}}/d$, we have

$$\mathbb{E}[\|\mathbf{g}_{k+1} - \nabla f(\mathbf{x}_{k+1})\|_2^2] \leq \frac{4LM^2\gamma^2\bar{\mathcal{E}}\eta^2}{B} + \frac{4M^2\bar{\mathcal{E}}}{B_0} \cdot \mathbb{1}(B_0 < n). \qquad \text{(C.13)}$$

Moreover, by Young's inequality, we have

$$\mathbb{E}[\|\mathbf{g}_{k+1}\|_2^2] \leq 2\mathbb{E}[\|\nabla f(\mathbf{x}_{k+1}) - \mathbf{g}_{k+1}\|_2^2] + 2\mathbb{E}[\|\nabla f(\mathbf{x}_k)\|_2^2]$$

$$\leq 2\mathbb{E}[\|\nabla f(\mathbf{x}_k)\|_2^2] + \frac{8LM^2\gamma^2\bar{\mathcal{E}}\eta^2}{B} + 4M^2\bar{\mathcal{E}} \cdot \mathbb{1}(B_0 < n).$$

Let $\eta^2 \leq 4^{-1}L^{-1}\gamma^{-2}$, it is evident that

$$\mathbb{E}[\|\mathbf{g}_{k+1}\|_2^2] \leq \frac{2M^2}{B}\bar{\mathcal{E}} + 2\mathbb{E}[\|\nabla f(\mathbf{x}_{k+1})\|_2^2] \leq 6M^2\bar{\mathcal{E}} + 2\mathbb{E}[\|\nabla f(\mathbf{x}_{k+1})\|_2^2],$$

which completes the induction for $\mathbf{g}_k$.

**Induction for $\mathcal{E}(\mathbf{x}_{k+1}, \mathbf{v}_{k+1})$:** By assumption (C.11), we have $\mathbb{E}[\|\mathbf{g}_k\|_2^2] \leq 6M^2\bar{\mathcal{E}} + 2\mathbb{E}[\|\nabla f(\mathbf{x}_k)\|_2^2] \leq 14M^2\bar{\mathcal{E}}$. Moreover, by (C.13) we have

$$\mathbb{E}[\|\mathbf{g}_{k+1} - \nabla f(\mathbf{x}_{k+1})\|_2^2] \leq \frac{4LM^2\gamma^2\bar{\mathcal{E}}\eta^2}{B} + \frac{4M^2\bar{\mathcal{E}}}{B_0} \cdot \mathbb{1}(B_0 < n).$$

Note that by Young's inequality $\|\mathbf{a} + \mathbf{b}\|_2^2 \leq 3\|\mathbf{a}\|_2^2/2 + 3\|\mathbf{b}\|_2^2$ we have

$$\mathcal{E}(\mathbf{x}, \mathbf{v}) \leq 5/2\|\mathbf{x}\|_2^2 + \frac{12}{\gamma^2}\|\mathbf{v}\|_2^2 + \frac{2uM}{\gamma^2}\left(3\|\mathbf{x}\|_2^2 + 6\|\mathbf{x}^*\|_2^2\right),$$

where we used the inequality

$$f(\mathbf{x}) - f(\mathbf{x}^*) \leq \frac{M}{2}\|\mathbf{x} - \mathbf{x}^*\|_2^2 \leq \frac{M}{4}\left(3\|\mathbf{x}\|_2^2 + 6\|\mathbf{x}^*\|_2^2\right).$$

Then if $\gamma^2 \leq 4Mu$, we have

$$\mathcal{E}(\mathbf{x}, \mathbf{v}) \leq \frac{12}{\gamma^2}\|\mathbf{v}\|_2^2 + \frac{16uM}{\gamma^2}\|\mathbf{x}\|_2^2 + \frac{12uM}{\gamma^2}\|\mathbf{x}^*\|_2^2. \tag{C.14}$$

Therefore, by (C.10) and the fact that $B_0 \geq \min\{1/\eta, n\}$, we have

$$\begin{aligned}
\mathbb{E}[\mathcal{E}(\mathbf{x}_{k+1}, \mathbf{v}_{k+1})] &\leq \mathbb{E}[\mathcal{E}(\mathbf{x}_k, \mathbf{v}_k)] - \frac{3um\eta}{\gamma}\mathbb{E}[\|\mathbf{x}_k\|_2^2] - \frac{2\eta}{\gamma}\mathbb{E}[\|\mathbf{v}_k\|_2^2] + \frac{32(\gamma^2 + u)uLM^2\bar{\mathcal{E}}\eta^3}{\gamma B} \\
&\quad + \frac{32(\gamma^2 + u)uM^2\eta^2}{\gamma^3}\bar{\mathcal{E}} + \frac{(288u + 14\gamma)M^2u\eta^2}{\gamma^2}\bar{\mathcal{E}} + \frac{16(d + b)u\eta}{\gamma} \\
&\leq \left(1 - \frac{\gamma m\eta}{6M}\right)\mathbb{E}[\mathcal{E}(\mathbf{x}_k, \mathbf{v}_k)] + \frac{(46\gamma^2 + 288u\gamma + 32u)M^2u\eta^2}{\gamma^3}\bar{\mathcal{E}} \\
&\quad + \frac{32(\gamma^2 + u)uLM^2\eta^3}{\gamma B}\bar{\mathcal{E}} + \frac{16(d + b)u\eta + 2um\|\mathbf{x}^*\|_2^2\eta}{\gamma},
\end{aligned}$$

where the last inequality follows from (C.14). We then set the step size as

$$\eta \leq \min\left\{\frac{\gamma^4 m}{48(46\gamma^2 + 288u\gamma + 32u)M^3u}, \frac{\gamma m^{1/2}}{48M^{3/2}(\gamma^2 + u)^{1/2}L^{1/2}}\right\},$$

and use the fact that $B \geq 1$, the following holds,

$$\begin{aligned}
\mathbb{E}[\mathcal{E}(\mathbf{x}_{k+1}, \mathbf{v}_{k+1})] &\leq \left(1 - \frac{\gamma m\eta}{6M}\right)\mathbb{E}[\mathcal{E}(\mathbf{x}_k, \mathbf{v}_k)] + \frac{\gamma m\eta}{24M}\bar{\mathcal{E}} + \frac{20(d + b)u\eta + 2um\|\mathbf{x}^*\|_2^2\eta}{\gamma} \\
&\leq \left(1 - \frac{\gamma m\eta}{8M}\right)\bar{\mathcal{E}} + \frac{16(d + b)u\eta + 2um\|\mathbf{x}^*\|_2^2\eta}{\gamma},
\end{aligned}$$

where the last inequality follows from the assumption that $\mathbb{E}[\mathcal{E}(\mathbf{x}_k, \mathbf{v}_k)] \leq \bar{\mathcal{E}}$. Since we have set

$$\bar{\mathcal{E}} = \mathcal{E}(\mathbf{x}_0, \mathbf{v}_0) + \frac{8M\left[16(d + b)u + um\|\mathbf{x}^*\|_2^2\right]}{\gamma^2 m} + \frac{G^2}{M^2},$$

it is evident that $\mathbb{E}[\mathcal{E}(\mathbf{x}_{k+1}, \mathbf{v}_{k+1})] \leq \bar{\mathcal{E}}$ holds as well. This completes the induction for $\mathcal{E}(\mathbf{x}_{k+1}, \mathbf{v}_{k+1})$.

$\square$

## C.2 Proof of Lemma B.3

*Proof.* Note that the semi-stochastic gradient $\mathbf{g}_k$ takes form

$$\mathbf{g}_k = \frac{1}{B} \sum_{i \in \mathcal{B}_k} \left( \nabla f_i(\mathbf{x}_k) - \nabla f_i(\mathbf{x}_{k-1}) \right) + \mathbf{g}_{k-1}.$$

By Lemma C.2 and (C.12), we have

$$\mathbb{E}[\|\mathbf{g}_k - \nabla f(\mathbf{x}_k)\|_2^2] \leq \frac{M^2}{B} \sum_{s=jm}^{jm+l} \mathbb{E}[\|\mathbf{x}_{s+1} - \mathbf{x}_s\|_2^2] + \frac{2}{B_0} \left( M^2 \mathbb{E}[\|\mathbf{x}_{jm}\|_2^2] + G^2 \right) \cdot \mathbb{1}(B_0 < n)$$

$$\leq \frac{LM^2}{B} \left( 2\eta^2 \mathbb{E}[\|\mathbf{v}_s\|_2^2] + \frac{u^2 \eta^4}{2} \mathbb{E}[\|\mathbf{g}_s\|_2^2] + \mathbb{E}[\|\boldsymbol{\epsilon}_s^x\|_2^2] \right)$$

$$+ \frac{2}{B_0} \left( M^2 \mathbb{E}[\|\mathbf{x}_{jm}\|_2^2] + G^2 \right) \cdot \mathbb{1}(B_0 < n).$$

Then by Lemma B.2 and (C.13), set $\eta^2 \leq \min\{\gamma^2/(6M^2 u^2), \gamma^2 \bar{\mathcal{E}}/(4G^2 u^2)\}$ and use the fact that $G^2 \leq M^2 \bar{\mathcal{E}}$, we obtain

$$\mathbb{E}[\|\mathbf{g}_k - \nabla f(\mathbf{x}_k)\|_2^2] \leq \frac{4LM^2 \gamma^2 \eta^2 \bar{\mathcal{E}}}{B} + \frac{4M^2 \bar{\mathcal{E}}}{B_0} \cdot \mathbb{1}(B_0 < n).$$

This completes the proof. $\qquad\qquad\square$

## C.3 Proof of Lemma B.4

*Proof.* Similar to the proof of Lemma B.2, we define

$$\mathcal{E}(\mathbf{x}, \mathbf{v}) = \|\mathbf{x}\|_2^2 + \|\mathbf{x} + 2\mathbf{v}/\gamma\|_2^2 + 8u\big(f(\mathbf{x}) - f(\mathbf{x}^*)\big)/\gamma^2.$$

Performing operator $\mathcal{L}$ on $\mathcal{A}(\mathbf{x}, \mathbf{v}) = e^{\lambda \mathcal{E}(\mathbf{x}, \mathbf{v})}$ gives

$$\mathcal{L}\mathcal{A} = \langle \nabla_{\mathbf{x}} \mathcal{A}, \mathbf{v} \rangle - \langle \nabla_{\mathbf{v}} \mathcal{A}, \gamma \mathbf{v} + u \nabla f(\mathbf{x}) \rangle + \langle \nabla_{\mathbf{v}}^2 \mathcal{A}, \gamma u \mathbf{I} \rangle$$

$$= \lambda \mathcal{A} \big( \langle \nabla_{\mathbf{x}} \mathcal{E}, \mathbf{v} \rangle - \langle \nabla_{\mathbf{v}} \mathcal{E}, \gamma \mathbf{v} + u \nabla f(\mathbf{x}) \rangle + \langle \nabla_{\mathbf{v}}^2 \mathcal{E}, \gamma u \mathbf{I} \rangle \big) + \lambda^2 \mathcal{A} \gamma u \|\nabla_{\mathbf{v}} \mathcal{E}\|_2^2$$

$$= \lambda \mathcal{A} \big( -4\|\mathbf{v}\|_2^2/\gamma - 4um\|\mathbf{x}\|_2^2/\gamma + 4u(2d+b)/\gamma \big) + \lambda^2 \mathcal{A} \gamma u \big\| 4\mathbf{x}/\gamma + 8\mathbf{v}/\gamma^2 \big\|_2^2$$

$$\leq \lambda \mathcal{A} \big( -4\|\mathbf{v}\|_2^2/\gamma - 4um\|\mathbf{x}\|_2^2/\gamma + 4u(2d+b)/\gamma \big) + \lambda^2 \mathcal{A} \gamma u \big( 32\|\mathbf{x}\|^2/\gamma^2 + 128\|\mathbf{v}\|^2/\gamma^4 \big\|_2^2,$$

where the last inequality is by Young's inequality. Let

$$\lambda \leq \min \left\{ \frac{\gamma^2}{64u}, \frac{m}{16} \right\},$$

we get

$$\mathcal{L}\mathcal{A} \leq \lambda \mathcal{A} \big( -2\|\mathbf{v}\|_2^2/\gamma - 2um\|\mathbf{x}\|_2^2/\gamma + 4u(2d+b)/\gamma \big). \qquad (C.15)$$

Moreover, by Young's inequality we have

$$\mathcal{E}(\mathbf{x}, \mathbf{v}) \leq 5/2\|\mathbf{x}\|_2^2 + \frac{12}{\gamma^2}\|\mathbf{v}\|_2^2 + \frac{2uM}{\gamma^2}\big(3\|\mathbf{x}\|_2^2 + 6\|\mathbf{x}^*\|_2^2\big).$$

where we use the inequality

$$f(\mathbf{x}) - f(\mathbf{x}^*) \leq \frac{M}{2}\|\mathbf{x} - \mathbf{x}^*\|_2^2 \leq \frac{M}{4}\big(3\|\mathbf{x}\|_2^2 + 6\|\mathbf{x}^*\|_2^2\big).$$

Assume $\gamma^2 \leq 4\mu M$, we have

$$\mathcal{E}(\mathbf{x}, \mathbf{v}) \leq \frac{12}{\gamma^2}\|\mathbf{v}\|_2^2 + \frac{16uM}{\gamma^2}\|\mathbf{x}\|_2^2 + \frac{12uM}{\gamma^2}\|\mathbf{x}^*\|_2^2. \qquad (C.16)$$

Plugging the above into (C.15) gives

$$\mathcal{L}\mathcal{A} \leq \lambda \mathcal{A} \left( -\frac{\gamma m}{8M} \mathcal{E} + 4u(2d+b)/\gamma + \frac{2um}{\gamma}\|\mathbf{x}^*\|_2^2 \right).$$

Therefore, we have the following for the Hamiltonian Langevin dynamics 1.1,

$$\frac{d\mathbb{E}[\mathcal{A}(\boldsymbol{X}_t, \boldsymbol{V}_t)]}{dt} = \mathbb{E}[\mathcal{L}\mathcal{A}(\boldsymbol{X}_t, \boldsymbol{V}_t)]$$

$$\leq \mathbb{E}\left[\mathcal{A}(\boldsymbol{X}_t, \boldsymbol{V}_t)\left(-\frac{\gamma m}{8M}\log\left(\mathcal{A}(\boldsymbol{X}_t, \boldsymbol{V}_t)\right) + \frac{4\lambda u(2d+b)}{\gamma} + \frac{2\lambda u m}{\gamma}\|\mathbf{x}^*\|_2^2\right)\right].$$

(C.17)

Note that $g(x) = x\log(x)$ is convex with respect to $x$, thus we have $\mathbb{E}[-A\log(A)] \leq -\log(\mathbb{E}[\mathcal{A}])\mathbb{E}[\mathcal{A}]$. Plugging this into (C.17) yields

$$\frac{d\mathbb{E}[\mathcal{A}]}{dt} \leq \mathbb{E}[\mathcal{A}]\left(-\frac{\gamma m}{8M}\log\left(\mathbb{E}[\mathcal{A}]\right) + \frac{4\lambda u(2d+b)}{\gamma} + \frac{2\lambda u m}{\gamma}\|\mathbf{x}^*\|_2^2\right),$$

(C.18)

where we abuse the notation $\mathcal{A}$ for simplification. Dividing $\mathbb{E}[\mathcal{A}]$ on both sides of (C.18) and rearranging terms give

$$\frac{d\log(\mathbb{E}[\mathcal{A}])}{dt} \leq -\frac{\gamma m}{8M}\log\left(\mathbb{E}[\mathcal{A}]\right) + \frac{4\lambda u(2d+b)}{\gamma} + \frac{2\lambda u m}{\gamma}\|\mathbf{x}^*\|_2^2.$$

This further lead to

$$\log(\mathbb{E}[\mathcal{A}(\boldsymbol{X}_t, \boldsymbol{V}_t)]) \leq \log(\mathbb{E}[\mathcal{A}(\boldsymbol{X}_0, \boldsymbol{V}_0)]) + \frac{16M\lambda u\left[4d + 2b + m\|\mathbf{x}^*\|_2^2\right)\right]}{\gamma^2 m}$$

$$= \lambda\mathcal{E}(\boldsymbol{X}_0, \boldsymbol{V}_0) + \frac{16M\lambda u\left[4d + 2b + m\|\mathbf{x}^*\|_2^2\right)\right]}{\gamma^2 m}.$$

(C.19)

Moreover, note that we have $\|\mathbf{a}\|_2^2 + \|\mathbf{b}\|_2^2 \geq \|\mathbf{a} - \mathbf{b}\|_2^2/2$, therefore,

$$\mathcal{E}(\mathbf{x}, \mathbf{v}) \geq \|\mathbf{x}\|_2^2/2 + \|\mathbf{v}/\gamma\|_2^2.$$

(C.20)

Let $\gamma < \sqrt{2}$, we have $\mathcal{E}(\mathbf{x}, \mathbf{v}) \geq (\|\mathbf{x}\|_2^2 + \|\mathbf{v}\|_2^2)/2$. Thus

$$\log\left(\mathbb{E}\left[e^{\lambda(\|\boldsymbol{X}_t\|_2^2 + \|\boldsymbol{V}_t\|_2^2)}\right]\right) \leq \log\left(\mathbb{E}\left[e^{2\lambda\mathcal{E}(\boldsymbol{X}_t, \boldsymbol{V}_t)}\right]\right)$$

$$\leq 2\lambda\mathcal{E}(\boldsymbol{X}_0, \boldsymbol{V}_0) + \frac{32M\lambda u\left[4d + 2b + m\|\mathbf{x}^*\|_2^2\right)\right]}{\gamma^2 m},$$

where the last inequality is obtained by replacing $\lambda$ with $2\lambda$ in (C.19), and thus we require $\lambda \leq \min\{\gamma^2/(128u), m/32\}$. This completes the proof. $\qquad\square$

## C.4   Proof of Lemma B.6

*Proof.* By Assumption 3.1, we have

$$f(\mathbf{x}) \leq f(\mathbf{x}^*) + \frac{M}{2}\|\mathbf{x} - \mathbf{x}^*\|_2^2 \leq f(\mathbf{x}^*) + M\|\mathbf{x}\|_2^2 + M\|\mathbf{x}^*\|_2^2,$$

where the second inequality is by Yong's inequality, which implies

$$f(\mathbf{x}) + u^{-1}\gamma^2\|\mathbf{x}\|_2^2/4 - f(\mathbf{x}^*) + M\|\mathbf{x}^*\|_2^2 \leq (M + u^{-1}\gamma^2/4)\|\mathbf{x}\|_2^2.$$

Divide both side by $(M + u^{-1}\gamma^2/4)/m$, and we have

$$\frac{m(f(\mathbf{x}) + u^{-1}\gamma^2\|\mathbf{x}\|_2^2/4 - f(\mathbf{x}^*) + -M\|\mathbf{x}^*\|_2^2)}{M + u^{-1}\gamma^2/4} \leq m\|\mathbf{x}\|_2^2.$$

According to Assumption 3.2, we have

$$\langle\nabla f(\mathbf{x}), \mathbf{x}\rangle \geq m\|\mathbf{x}\|_2^2 - b \geq \frac{m(f(\mathbf{x}) + u^{-1}\gamma^2\|\mathbf{x}\|_2^2/4}{M + u^{-1}\gamma^2/4} - \frac{f(\mathbf{x}^*) + M\|\mathbf{x}^*\|_2^2)}{M + u^{-1}\gamma^2/4} - b,$$

which directly completes the proof by dividing both side by 2. $\qquad\square$

# D  Proof of additional lemmas

In this section we prove the additional supporting lemmas.

## D.1  Proof of Lemma C.1

*Proof.* Applying Assumption 3.1 and noting that $\mathbf{x}_0 = \mathbf{0}$, we have

$$\|\nabla f_i(\mathbf{x})\|_2 \leq \|\nabla f_i(\mathbf{x}_0)\|_2 + M\|\mathbf{x} - \mathbf{x}_0\|_2 = \|\nabla f_i(\mathbf{0})\|_2 + M\|\mathbf{x}\|_2.$$

By setting $G = \max_{i \in [n]} \|\nabla f_i(\mathbf{0})\|_2$, we complete the proof. $\qquad\square$

## D.2  Proof of Lemma C.2

*Proof.* By the formula of $\mathbf{g}_k$, and let $k = jm + l$ denote the $l$-th iterate in the $j$-th epoch of Algorithm 1, we have

$$\mathbb{E}[\|\mathbf{g}_{k+1} - \nabla f(\mathbf{x}_{k+1})\|_2^2] = \mathbb{E}\left[\left\|\frac{1}{B}\left(\sum_{i \in \mathcal{B}_{k+1}}\left[\nabla f_i(\mathbf{x}_{k+1}) - \nabla f_i(\mathbf{x}_k)\right]\right) + \mathbf{g}_k - \nabla f(\mathbf{x}_{k+1})\right\|_2^2\right]$$

$$= \mathbb{E}\left[\left\|\frac{1}{B}\left(\sum_{i \in \mathcal{B}_{k+1}}\left[\nabla f_i(\mathbf{x}_{k+1}) - \nabla f_i(\mathbf{x}_k)\right]\right) - \left(\nabla f(\mathbf{x}_{k+1}) - \nabla f(\mathbf{x}_k)\right)\right\|_2^2\right]$$

$$+ \mathbb{E}[\|\mathbf{g}_k - \nabla f(\mathbf{x}_k)\|_2^2].$$

By Lemma A.1 in [36], we know that

$$\mathbb{E}\left[\left\|\frac{1}{B}\left(\sum_{i \in \mathcal{B}_{k+1}}\left[\nabla f_i(\mathbf{x}_{k+1}) - \nabla f_i(\mathbf{x}_k)\right]\right) - \left(\nabla f(\mathbf{x}_{k+1}) - \nabla f(\mathbf{x}_k)\right)\right\|_2^2\right]$$

$$\leq \frac{1}{B}\mathbb{E}[\|\nabla f_i(\mathbf{x}_{k+1}) - \nabla f_i(\mathbf{x}_k)\|_2^2].$$

Thus, it follows that

$$\mathbb{E}[\|\mathbf{g}_{k+1} - \nabla f(\mathbf{x}_{k+1})\|_2^2] \leq \frac{1}{B}\mathbb{E}[\|\nabla f_i(\mathbf{x}_{k+1}) - \nabla f_i(\mathbf{x}_k)\|_2^2] + \mathbb{E}[\|\mathbf{g}_k - \nabla f(\mathbf{x}_k)\|_2^2]$$

$$\leq \frac{1}{B}\sum_{s=jm}^{jm+l}\mathbb{E}[\|\nabla f_i(\mathbf{x}_{s+1}) - \nabla f_i(\mathbf{x}_s)\|_2^2] + \mathbb{E}[\|\mathbf{g}_{jm} - \nabla f(\mathbf{x}_{jm})\|_2^2]$$

$$\leq \frac{M^2}{B}\sum_{s=jm}^{jm+l}\mathbb{E}[\|\mathbf{x}_{s+1} - \mathbf{x}_s\|_2^2] + \frac{1}{B_0}\mathbb{E}[\|\nabla f_i(\mathbf{x}_{jm})\|_2^2] \cdot \mathbb{1}(B_0 < n)$$

$$\leq \frac{M^2}{B}\sum_{s=jm}^{jm+l}\mathbb{E}[\|\mathbf{x}_{s+1} - \mathbf{x}_s\|_2^2] + \frac{2}{B_0}\mathbb{E}[\|\mathbf{x}_{jm}\|_2^2 + G^2] \cdot \mathbb{1}(B_0 < n),$$

where the first inequality is by Young's inequality, the second inequality is by Assumption 3.1, the third inequality follows Lemma A.1 in [36], the last inequality is by Lemma C.1. This completes the proof. $\qquad\square$

# E  Additional experimental results

In this section, we provide additional experimental results.

## E.1  Comparison of posterior distributions

Here we conduct additional comparison in terms of sampled posterior distributions for ICA. In detail, we use HMC with metropolis hasting correction to generate the ground truth. Similar to [23], we randomly choose two variables ($W_{1,1}$ and $W_{5,17}$) from the parameter matrix $\mathbf{W}$ and display their

Table 2: Summary of datasets for Bayesian logistic classification.

| Dataset | *pima* | *a3a* | *mushroom* | *a9a* |
|---|---|---|---|---|
| $n$ (training) | 600 | 3185 | 6000 | 32,561 |
| $n$ (test) | 168 | 29376 | 2124 | 16281 |
| $d$ | 8 | 123 | 122 | 123 |

marginal distributions after 1000 data passes in Figures 4(a)-4(f) (row 1) and Figures 4(g)-4(l) (row 2) respectively. It can be observed that the proposed SRVR-HMC (as well as SVRG-LD and SVR-HMC) can well approximate the ground truth, while SGLD and SGHMC cannot provide accurate approximation. This further validates the superior performance of SRVR-HMC and other variance reduced algorithms (SVRG-LD, SVR-HMC).

(a) SGLD (b) SGHMC (c) SUL-MCMC (d) SVRG-LD (e) SVR-HMC (f) SRVR-HMC

(g) SGLD (h) SGHMC (i) SUL-MCMC (j) SVRG-LD (k) SVR-HMC (l) SRVR-HMC

Figure 4: Marginal distributions of the posterior samples generated by Langevin dynamics based algorithms (red line) including SGLD, SGHMC, SG-UL-MCMC, SVRG-LD, SVR-HMC and SRVR-HMC, as well as the ground truth (green line). (Here we use SUL-MCMC to denote SG-UL-MCMC due to the space limit.)

## E.2   ICA with larger dataset

We also ran additional experiments for ICA on a larger dataset (extract a larger subset from the original dataset, i.e., $n = 10000$), which is displayed in Figure 5. It can be seen that the proposed SRVR-HMC algorithm achieves the best performance among all methods.

## E.3   Bayesian Logistic Regression

Assume we are given data $\{\mathbf{x}_i, y_i\}_{i=1,\dots,n}$ where $\mathbf{x}_i$ denotes the feature vector and $y_i \in \{-1, 1\}$ denotes the corresponding label. Then the probability density function of the label $y$ given the feature $\mathbf{x}_i$ and model vector $\boldsymbol{\beta}$ is modeled as $p(y|\mathbf{x}_i, \boldsymbol{\beta}) = 1/(1 + e^{-y_i \boldsymbol{\beta}^\top \mathbf{x}_i})$. We further assume the model vector $\boldsymbol{\beta}$ follows a Gamma prior $p(\boldsymbol{\beta}) \propto \|\boldsymbol{\beta}\|_2^{-\lambda} \exp(-\theta\|\boldsymbol{\beta}\|_2)$, where $\lambda$ and $\boldsymbol{\beta}$ are fixed parameter. In the Bayesian logistic classification task, we aim to sample the posterior distribution

$$p(\boldsymbol{\beta}|\{\mathbf{x}_i, y_i\}_{i=1,\dots,n}) = p(\boldsymbol{\beta}) \prod_{i=1}^{n} p(y_i|\mathbf{x}_i, \boldsymbol{\beta}).$$

Let $f(\boldsymbol{\beta}) = -\log p(\boldsymbol{\beta}|\{\mathbf{x}_i, y_i\}_{i=1,\dots,n})$. Each function $f_i(\boldsymbol{\beta})$ in (1.3) takes the form of $f_i(\boldsymbol{\beta}) = -n \log(p(y_i|\mathbf{x}_i, \boldsymbol{\beta})) + \lambda \log(\|\boldsymbol{\beta}\|_2) + \theta\|\boldsymbol{\beta}\|_2$.

Figure 5: Results for ICA on a larger dataset (training sample size: $n = 10000$). $X$-axis represents the number of data passes and $Y$-axis represents the negative log likelihood on the test dataset.

Figure 6: Comparison of different algorithms for Bayesian logistic regression, where Y-axis denotes the negative log likelihood on test datasets and X-axis denotes number of data passes.

We compare the performance of the proposed algorithm with SGLD [50], SGHMC [16], SG-UL-MCMC [18], SVRG-LD [25], and SVR-HMC [55] on *pima*, *a3a*, *mushroom*, and *a9a* dataset, which are available in UCI[5] [11] and LibSVM[6] [38] libraries. We summarize the detail of these datasets in Table 2. We run all algorithms on the training dataset, where the hyper parameters are tuned under the guidance of their theory. Moreover, we compute the sample path average of the position variable as the output. Then, such output is applied to conduct classification tasks on the test datasets, and we plot the negative log likelihood in Figures 6(a) - 6(d). It can be seen that the proposed algorithm SRVR-HMC outperforms all baseline algorithms on these four dataset, which is consistent with our theory.