[Reviews · NeurIPS 2019]

Reviewer 1



Their algorithm is clearly explained. It is easy to implement and have good theoretical convergence properties. As the authors have pointed out, a closely related (but not identical) approach is already used in literature for an optimisation problem however its use in sampling is novel. I find this paper a high quality work which potentially can have some impact both in terms of the practical applications of the proposed sampler as well as the way the convergence analysis is carried out.

Reviewer 2



UPDATE: The authors' rebuttal addressed some of my concerns, while this work still feels incremental, I'm inclined to accept it now. Strength - A theoretical analysis of the convergence rate for the proposed algorithm. - A nice review of gradient complexity for existing variants of SG-HMC and SGLD. - Experiments demonstrate improved mixing for the proposed algorithm over competing methods. Weakness - The proposed algorithm is a relatively standard extension of SG-HMC and SGLD. - While the proposed framework and analysis and bounds are a contribution, such results are fairly familiar in the literature. References: The following articles might also be related: Bardenet, R., Doucet, A., & Holmes, C. (2017). On Markov chain Monte Carlo methods for tall data. The Journal of Machine Learning Research, 18(1), 1515-1557. Brosse, N., Durmus, A., & Moulines, E. (2018). The promises and pitfalls of stochastic gradient Langevin dynamics. In Advances in Neural Information Processing Systems (pp. 8268-8278). Dang, K. D., Quiroz, M., Kohn, R., Tran, M. N., & Villani, M. (2018). Hamiltonian Monte Carlo with energy conserving subsampling. arXiv preprint arXiv:1708.00955v2.

Reviewer 3



Update: The authors have helpfully pointed out that they do provide some guidelines on setting the hyperparameters. This paper creatively combines underdamped Langevin MCMC work of Cheng et al. with the gradient estimator SPIDER of Fang et al. This allows the paper to use the theoretical result from Fang et al. to prove a better bound for achieving epsilon in 2-Wasserstein distance. Effectively it is the UL-MCMC algorithm with a better gradient estimator. This isn't meant to imply that the work is trivial as adapting any insight from the optimisation literature for use in a HMC algorithm requires careful work to yield measurable improvements. The paper was technically sound with a compelling theoretical analysis and adequate experimental results. The theory already suggests the algorithm works best on a small batch size, but it's unclear if there is any way to do anything other than roughly estimate how that hyperparameter should be set. The paper is wonderfully organised though also a fairly dense read. This is likely unavoidable given the theoretical contributions. The supplementary section greatly helped in understanding the paper, but the key ideas needed were still in the main paper. This paper is a great step forward in incorporating SPIDER and it will be nice to continue to see more variance reduction methods be introduced for Langevin methods, where they are sorely needed.

[Author Response · NeurIPS 2019]

**Response to Reviewer 1:** Thank you for your supportive comments! We will fix the typos in the final version.

**Response to Reviewer 2:** Thanks for your helpful comments.

**Q1:** "The proposed algorithm is a relatively standard extension of SG-HMC and SGLD. While the proposed framework
and analysis and bounds are a contribution, such results are fairly familiar in the literature."
**A1:** From the perspective of the design of our algorithm, we admit that our algorithm is an extension of SG-HMC.
While SG-HMC and SGLD type algorithms have been widely studied in the literature, and might look familiar to you,
our proposed algorithm is new, and its theoretical analysis has never been done in the literature and therefore is also
new. More importantly, the corresponding theoretical guarantees of our algorithm outperform the state-of-the-art. This
is also verified by our experiments.
**Q2:** "The following articles might also be related..."
**A2:** Thank you for pointing out these related articles. We will definitely cite and discuss them in the final version.
**Q3:** "Why not show figures that compare these samples against some ground truth, for example, those obtained by
HMC (which is feasible to obtain for GMM and ICA)? See those of ...."
**A3:** Thank you for your suggestion to show the comparison between the samples generated by our method and the
ground truth obtained by HMC. We have added this additional comparison in Figure 1 for ICA. In detail, we use HMC
with metropolis hasting correction to generate the ground truth. Following the references pointed out by you, we
randomly choose two variables ($W_{1,1}$ and $W_{5,17}$) from the parameter matrix $\mathbf{W}$ and display their marginal distributions
after 1000 data passes in Figures 1(a)-1(f) (row 1) and Figures 1(g)-1(l) (row 2) respectively. It can be observed that the
proposed SRVR-HMC (as well as SVRG-LD and SVR-HMC) can well approximate the ground truth, while SGLD and
SGHMC cannot provide accurate approximation. This further validates the superior performance of SRVR-HMC and
other variance reduced algorithms (SVRG-LD, SVR-HMC). We will add these experimental results in the final version.

| (a) SGLD | (b) SGHMC | (c) SG-UL-HMC | (d) SVRG-LD | (e) SVR-HMC | (f) SRVR-HMC |
| --- | --- | --- | --- | --- | --- |
| (g) SGLD | (h) SGHMC | (i) SG-UL-MCMC | (j) SVRG-LD | (k) SVR-HMC | (l) SRVR-HMC |

Figure 1: Marginal distributions of the posterior samples generated by Langevin dynamics based algorithms (red line)
including SGLD, SGHMC, SG-UL-MCMC, SVRG-LD, SVR-HMC and SRVR-HMC, as well as the ground truth
(green line).

**Response to Reviewer 3:** Thank you for your supportive comments.

**Q1:** "It's unclear if there is any way to do anything other than roughly estimate how
that hyperparameter should be set."
**A1:** As suggested in Corollary 3.5 of our paper, the batch size parameter $B$ should be
smaller than $O(\epsilon^{-2}\mu_*^{-1/2} \wedge \sqrt{n})$, where $\epsilon$ is the precision parameter, $\mu_*$ is the spectral
gap and $n$ is the sample size of the dataset. For general non-log-concave distributions,
the spectral gap $\mu_*$ can be very small, thus we often have $\epsilon^{-2}\mu_*^{-1/2} \geq \sqrt{n}$. Therefore,
the batch size parameter in our experiments is chosen to guarantee that $B \leq C\sqrt{n}$ with
$C$ being a tuning parameter. This is validated by the sensitivity analysis of batch size
$B$ in Figures 2(b), 3(c) and 3(d) in our submission.
**Q2:** "The paper could have been improved a bit by maybe running the algorithm on
slightly larger datasets."
**A2:** We indeed ran the algorithm on a large dataset (a9a, training sample size: 32561,
test sample size: 16281) for Bayesian logistic regression. Due to the space limit, we put
it in Appendix E. Follow your suggestion, we also ran additional experiments for ICA
on a larger dataset (extract a larger subset from the original dataset, i.e., $n = 10000$),
which is displayed in Figure 2. It can be seen that the proposed SRVR-HMC algorithm
achieves the best performance among all methods. We will add more experiments on
larger datasets in the appendix of the final version.

Figure 2: Results for ICA on a larger dataset (training sample size: $n = 10000$). $X$-axis represents the number of data passes and $Y$-axis represents the negative log likelihood on the test dataset.

[Meta-Review · NeurIPS 2019]

The reviewers expressed concern on the novelty of the algorithm, but still think it is well written and should be accepted for NIPS. We encourage the authors to carefully revise the work for the final version.